# Maternal polycystic ovary syndrome and Offspring's Risk of Cardiovascular diseases in Childhood and Young Adulthood

Fen Yang [1,2,13] ✉, Ziliang Wang [3,4,13], Henrik Toft Sørensen [5], Imre Janszky[1,6], Mika Gissler [7,8,9], Wei Yuan[3], Maohua Miao[3], Nathalie Roos[10], Anna-Karin Wikström[11], Jiong Li[4] & Krisztina D. László[1,12]

Children born to mothers with polycystic ovary syndrome have a higher prevalence of cardiovascular risk factors and of subclinical cardiovascular disease, but the association between maternal polycystic ovary syndrome and cardiovascular disease in offspring is unclear. We conduct a register-based cohort study of 6 839 703 live singleton births from Denmark (1973–2016) and Sweden (1973–2014) and follow them for up to 48 years. Using Cox regression models, we find that offspring of mothers with polycystic ovary syndrome have a higher risk of overall cardiovascular diseases and of its specific subtypes, independently of comorbidities related to polycystic ovary syndrome. Cousin analyzes suggest that familial confounding does not explain our results. If our findings are replicated by future studies, children of women with polycystic ovary syndrome may benefit from early cardiovascular prevention efforts.

Polycystic ovary syndrome (PCOS) is one of the most common endocrine disorders in women of reproductive age, and its estimated prevalence ranges from 10% to 13%[1]. According to the latest International Evidence-based Guideline for the Assessment and Management of PCOS[1,2], PCOS is a heterogeneous condition characterized by several reproductive, cardiometabolic, and psychological disorders, including infertility, obesity, diabetes, cardiovascular disease (CVD), depression, and anxiety[3]. When women with PCOS become pregnant, they are at increased risk of pregnancy complications and adverse pregnancy outcomes, such as gestational diabetes, gestational hypertension, preeclampsia, cesarean section, and preterm delivery[4–6].

For the fetus, maternal PCOS per se and the associated comorbidities and pregnancy complications may produce a suboptimal intrauterine environment, which in turn may adversely influence fetal growth and long-term health after birth[7]. The few long-term studies on the health of offspring exposed to maternal PCOS focused mainly on reproductive, metabolic or neurodevelopmental outcomes[7,8]. Children of mothers with PCOS are more likely to have subclinical CVD markers including higher pulse pressure and carotid intima-media thickness than the unexposed[9]. However, knowledge regarding the link between maternal PCOS and clinical CVD in offspring is limited. Only two earlier studies investigated this possible association; both reported a 27 to

[1]Department of Global Public Health, Karolinska Institutet, Stockholm, Sweden. [2]Institute of Environmental Medicine, Karolinska Institutet, Stockholm, Sweden. [3]Shanghai-MOST Key Laboratory of Health and Disease Genomics, NHC Key Lab of Reproduction Regulation, Shanghai Institute for Biomedical and Pharmaceutical Technologies, Shanghai, China. [4] Department of Clinical Medicine and Department of Clinical Epidemiology, Aarhus University Hospital, Aarhus, Denmark. [5]Department of Clinical Epidemiology, Center for Population Medicine, Aarhus University and Aarhus University Hospital, Aarhus, Denmark. [6]Department of Public Health and Nursing, Norwegian University of Science and Technology, Trondheim, Norway. [7]Department of Data and Analytics, Finnish Institute for Health and Welfare, Helsinki, Finland. [8]Academic Primary Health Care Centre, Region Stockholm, Stockholm, Sweden. [9]Department of Molecular Medicine and Surgery, Karolinska Institutet, Stockholm, Sweden. [10]Division of Clinical Epidemiology, Department of Medicine Solna, Karolinska University Hospital and Karolinska Institutet, Stockholm, Sweden. [11]Department of Women's and Children's Health, Uppsala University, Uppsala, Sweden. [12]Department of Public Health and Caring Sciences, Uppsala University, Uppsala, Sweden. [13]These authors contributed equally: Fen Yang, Ziliang Wang. ✉e-mail: fen.yang@ki.se

31% increased risk of hospitalization due to CVD in offspring born to mothers with PCOS[10,11], but with wide confidence intervals due to rare CVD events in children.

In this work, involving almost seven million births from Denmark and Sweden, we examine the association between maternal PCOS and the risk of CVD in offspring. In addition, as there is a familial predisposition to cardiometabolic diseases in offspring born to women with PCOS[12], we use a cousin comparison design to assess whether familial genetic or environmental characteristics contribute to the association. We show that individuals born to mothers with PCOS have increased risks of CVD in childhood and young adulthood, and that this cannot be explained by familial confounding, PCOS-related comorbidities, or adverse pregnancy outcomes.

## Results

In the cohort of 6 839 703 singletons, 51 723 (0.76%) were exposed to maternal PCOS. Compared to their unexposed counterparts, offspring of mothers with PCOS were more likely to be born after 2003, with a preterm or large for gestational age (LGA) birth. Their mothers were more likely to be foreign-born, to have higher education, to be single, non-smoking, nulliparous, and obese, to have undergone assisted reproductive treatment (ART), and to have had a diagnosis of diabetes, hypertensive disease, or psychiatric disorders before childbirth (Table 1).

### Maternal PCOS and risks of overall CVD and its main types

During the up to 48 years of follow-up, 384 274 offspring (5.6%) had a diagnosis of CVD. We observed consistently higher incidence rates of overall CVD in offspring exposed to maternal PCOS than in the group of unexposed offspring across the entire follow-up period (Supplementary Fig. 1). Adjustment for potential confounders did not substantially change the estimates. Compared with unexposed offspring, those exposed to maternal PCOS had higher risks of overall CVD (hazard ratio (HR) in the fully adjusted model, 1.21, 95% confidence intervals (CIs), 1.15 to 1.27) and of some CVD subtypes, including ischemic heart disease (IHD), acute myocardial infarction (AMI), and hypertensive disease; the point estimate corresponding to the risk of stroke in offspring according to exposure to maternal PCOS was also increased, although the 95% CIs included one (Table 2). We found similar associations between maternal PCOS and overall CVD diagnosed among offspring in the following age groups: <10 years, 10–19 years, 20–29 years, 30–39 years, and ≥40 years (Supplementary Table 1).

### The joint effect of maternal PCOS and comorbidities

Offspring of mothers who had both PCOS and comorbidities (including diabetes, hypertensive disease, or psychiatric disorders) had higher CVD risk than those whose mothers had only PCOS (Table 3). The synergy index for the interaction between PCOS and all comorbidities suggested that the effects of maternal PCOS and all comorbidities on the risk of CVD in the offspring are super-additive.

### Cousin analysis

In the cousin analysis, most associations were attenuated, but the association of maternal PCOS with the risk of overall CVD, IHD, AMI, and hypertensive disease persisted. The hazard ratios for IHD and AMI in the cousin analysis were greater than the corresponding hazard ratios in the main analysis (Table 4).

### Mediation by preterm birth, small for gestational age (SGA), or LGA birth, congenital heart disease (CHD), and diabetes

The association between maternal PCOS and risk of CVD was largely independent of preterm birth, SGA or LGA birth, CHD, or diabetes. Congenital heart disease mediated 10% of the observed association, but there was limited evidence for mediation in the case of the other four conditions (Table 5).

### Sensitivity analyzes

The association between maternal PCOS and overall CVD risk in offspring did not differ by offspring's sex, country of birth, maternal use of ART, or the year of maternal PCOS diagnosis (Supplementary Table 2). The results did not substantially change when we (1) repeated analyzes in the propensity-score-matched (PSM) sub-cohort (Supplementary Table 3); (2) adjusted for maternal smoking or body-mass index (BMI) in early pregnancy, in addition to factors included in the main model; (3) restricted analyzes to the years when specific outpatient clinic data were available; (4) restricted our definition of CVD to cases identified only from the national patient registers (n = 383 030); or (5) repeated the main analysis with imputed data for covariates with missing information (Supplementary Table 2).

## Discussion

We found that maternal PCOS was associated with increased risks of overall CVD and major subtypes of CVD in offspring. The association between PCOS and CVD persisted from childhood to early middle-age and was not fully explained by maternal comorbidities, ART, preterm birth, or abnormal fetal growth. The comparable hazard ratios for overall CVD and several CVD types, including IHD, AMI, and hypertensive disease, from the main analysis with those in the cousin comparison analysis suggest that shared familial factors may not fully account for the association between maternal PCOS and the risk of CVD in offspring.

To our knowledge the association between exposure to maternal PCOS and the offspring's risk of developing CVD has been investigated only in two earlier studies, one conducted in Australia (N = 28 226) and one in Canada (N = 1 038 375)[10,11]. In line with our findings, these studies reported that offspring exposed to maternal PCOS have an increased risk of hospitalization for CVD compared to the unexposed. However, the short follow-up period, i.e., 30 years in the Australian and 13 years in the Canadian study, did not permit to investigate the association between maternal PCOS and the offspring's CVD risk in older ages. None of the studies could control for familial genetic or environmental factors that may contribute to the risk of both maternal PCOS and CVD among offspring. Our study extends the limited evidence in this field in multiple ways. Our large study population and the almost five decades follow-up allowed us to detect a modest association between maternal PCOS and increased risks both of total CVD and specific CVD subtypes, to include outcomes from childhood to early middle-age, to control for a large range of confounders in multivariable models and using a family study design and to investigate effect modification by comorbidity and demographic factors and mediation by birth outcomes in these associations.

An important challenge in this research area is to examine the effect of maternal PCOS alone, since mothers with PCOS tend to have comorbidities (such as obesity, diabetes, hypertension, and psychiatric disorders)[3] or to seek ART to conceive[13], factors which in turn are linked with their children's CVD risk later in life[14–18]. We found that the association between maternal PCOS and increased risk of CVD persisted even among offspring of women without these comorbidities or ART, suggesting that these conditions could not fully explain our findings. However, when we investigated the joint effect of PCOS and comorbidities, we observed synergism, i.e., the risks of CVD were higher among offspring of women with both PCOS and a comorbidity than would be expected from the effects of each exposure alone. In other words, the adverse effects of these comorbidities on cardiovascular risk were more pronounced among women with PCOS than among women without PCOS. Further research is necessary to elucidate the mechanisms driving this synergism. Nevertheless, these findings may highlight the importance of proactive clinical

**Table 1 | Baseline Characteristics of the Study Population**

| | No maternal PCOS (N = 6 787 980) | Maternal PCOS (N = 51 723) |
|---|---|---|
| **Characteristics of offspring** | | |
| **Study country, n (%)** | | |
| Denmark | 2 648 462 (39.0) | 20 235 (39.1) |
| Sweden | 4 139 518 (61.0) | 31 488 (60.9) |
| **Calendar year of birth, n (%)** | | |
| 1973–1978 | 990 991 (14.6) | 523 (1.0) |
| 1979–1984 | 864 661 (12.7) | 1 088 (2.1) |
| 1985–1990 | 972 874 (14.3) | 2 054 (4.0) |
| 1991–1996 | 1 034 976 (15.3) | 3 971 (7.7) |
| 1997–2002 | 887 817 (13.1) | 6 428 (12.4) |
| 2003–2008 | 951 753 (14.0) | 13 267 (25.7) |
| 2009–2016 | 1 084 908 (16.0) | 24 392 (47.2) |
| **Male, n (%)** | 3 487 321 (51.4) | 26 683 (51.6) |
| **Congenital heart disease, n (%)** | 100 170 (1.5) | 1 243 (2.4) |
| **Preterm birth, n (%)** | 310 119 (4.6) | 3 683 (7.1) |
| **Small for gestational age, n (%)** | 634 672 (9.3) | 4 304 (8.3) |
| **Large for gestational age, n (%)** | 633 386 (9.3) | 6 823 (13.2) |
| **Maternal characteristics** | | |
| **Country of origin same as the study country, n (%)** | 5 833 226 (85.9) | 41 170 (79.6) |
| **Age at the time of the index birth (years), n (%)** | | |
| ≤19 | 203 895 (3.0) | 1 488 (2.9) |
| 20–24 | 1 358 370 (20.0) | 9 699 (18.8) |
| 25–29 | 2 405 425 (35.4) | 17 895 (34.6) |
| 30–34 | 1 908 878 (28.1) | 15 671 (30.3) |
| ≥35 | 911 412 (13.4) | 6 970 (13.5) |
| **Level of education at the time of the index birth, n (%)** | | |
| Primary and lower secondary | 1 494 582 (22.0) | 9 931 (19.2) |
| Upper secondary | 3 062 090 (45.1) | 22 076 (42.7) |
| Bachelor or higher | 2 114 942 (31.2) | 19 240 (37.2) |
| Unknown | 116 366 (1.7) | 476 (0.9) |
| **Being married/registered partnership at the time of the index birth, n (%)** | 3 503 391 (51.6) | 24 860 (48.1) |
| **Parity at the time of the index birth, n (%)** | | |
| 1 | 2 945 493 (43.4) | 28 453 (55.0) |
| 2 | 2 496 503 (36.8) | 16 924 (32.7) |
| ≥3 | 1 345 984 (19.8) | 6 346 (12.3) |
| **Smoking in early pregnancy, n (%)** | 792 278 (16.3) | 5 953 (12.0) |
| **Body-mass index in early pregnancy (kg/m², n (%))** | | |
| <18.5 | 128 808 (3.15) | 739 (1.64) |
| 18.5–24.9 | 2 013 501 (49.3) | 16 424 (36.4) |
| 25.0–29.9 | 768 683 (18.8) | 11 850 (26.3) |
| ≥30.0 | 323 297 (7.91) | 11 264 (25.0) |
| Unknown | 854 281 (20.9) | 4 806 (10.7) |
| **Assisted reproductive treatment, n (%)** | 94 478 (2.9) | 7 869 (17.1) |
| **Diabetes before delivery, n (%)** | 100 147 (1.5) | 3 263 (6.3) |
| **Hypertensive disease before delivery, n (%)** | 233 635 (3.4) | 3 944 (7.6) |
| **Psychiatric disorders before delivery, n (%)** | 292 070 (4.3) | 5 919 (11.4) |
| **Family history of cardiovascular disease, n (%)** | 3 087 825 (45.5) | 23 047 (44.6) |

Abbreviations: *PCOS* polycystic ovary syndrome, *n* number.

management, particularly for pregnant women with multiple complications. Early screening and preventive measures among children born to mothers with PCOS and comorbidities may have the potential to mitigate their long-term cardiovascular consequences.

Our findings of no sex differences in the association between maternal PCOS and CVD risk in offspring is consistent with results of several earlier studies suggesting that the effects of polygenic risk of PCOS on cardiometabolic health were similar in men and women[12]. However, other studies suggest the potential for a sex-specific effect; some studies suggest that males have a greater cardiovascular risk compared to females at the same level of genetic predisposition for PCOS[19], while others have observed altered cardiometabolic features predominantly among girls born to women with PCOS[7,20]. Further research is needed to elucidate these sex differences.

There are several plausible explanations for our findings. First, shared genetic predisposition or environmental risk factors, such as maternal lifestyle and socioeconomic status, may contribute to the observed association between maternal PCOS and CVD risk in offspring[21,22]. However, we observed no substantial changes in the associations after adjusting for several maternal characteristics or performing cousin-comparison analyzes. Nevertheless, the role of genetic confounding cannot be excluded in the present study. Furthermore, daughters of women with PCOS are at an increased risk of developing PCOS, which, in turn, could be associated with subsequent CVD risk[3]. Unfortunately, we could not investigate the role of offspring's PCOS in the association between maternal PCOS and offspring's CVD. The reason is that a high proportion of the female offspring were too young to receive a PCOS diagnosis, as symptoms typically emerge post-puberty. Additionally, some offspring without a PCOS diagnosis during our study period may receive a diagnosis in the future, beyond our observation period and the period of data availability. The potential misclassification of PCOS among offspring could lead to underestimation of its mediated effect. Second, the hyperandrogenic and hyper-insulinemic intrauterine environment created by PCOS may influence cardiovascular health through fetal programing[23]. A suboptimal intrauterine environment characterized by inflammation, oxidative stress, and placental dysfunction[24] could lead to abnormal fetal growth, preterm birth or diabetes, which in turn increase the risk of CVD in later life[25,26]. Our mediation analysis suggested that the associations between maternal PCOS and CVD in offspring was more likely mediated by CHD than by preterm birth and abnormal fetal growth, suggesting that PCOS may involve developmental cardiac programming through cardiovascular remodeling and dysfunction other than prematurity and extreme fetal growth. The potential mediating role of CHD could be supported by previous findings that: 1) maternal PCOS is associated with an increased risk of CHD in offspring[10], and 2) CHD is a major cause of CVD in childhood and early adulthood due to its influence on anatomic and hemodynamic abnormalities in the cardiovascular system[27]. Similarly, we found no evidence of mediation by diabetes. A further hypothesis is that *in utero* hyperandrogenemia directly leads to accelerated atherosclerosis progression in the fetus[28]. Previous findings that offspring exposed to maternal PCOS had higher blood pressure and intima-media thickness[7,9] and our findings concerning associations of PCOS with increased risk of AMI, IHD, and ischemic stroke, all of which involve atherosclerosis progression, supported this plausible mechanism. Maternal PCOS may also increase the offspring's risk of CVD by increasing the risk of obesity in the offspring, which in turn is a well-known cardiovascular risk factor. Regrettably, information on offspring's BMI was not available in registers, preventing us from further investigating this hypothesis.

## Limitations

The study had several limitations. First, the prevalence of maternal PCOS in our cohort is lower than that reported in the general

**Table 2 | Hazard ratios and 95% confidence intervals for overall and specific cardiovascular diseases in offspring according to maternal polycystic ovary syndrome**

| Exposure | Number of events | Event rate, per 10 000 person-years | HR (95% CI) | | | |
|---|---|---|---|---|---|---|
| | | | Model 1[a] | Model 2[b] | Model 3[c] | Model 4[d] |
| **Overall CVD** | | | | | | |
| No PCOS | 382 782 | 23.93 | 1.0 (Reference) | 1.0 (Reference) | 1.0 (Reference) | 1.0 (Reference) |
| PCOS | 1 492 | 22.41 | 1.59 (1.51–1.67) | 1.23 (1.17–1.29) | 1.53 (1.45–1.61) | 1.21 (1.15–1.27) |
| **Ischemic heart disease** | | | | | | |
| No PCOS | 8 640 | 0.53 | 1.0 (Reference) | 1.0 (Reference) | 1.0 (Reference) | 1.0 (Reference) |
| PCOS | 20 | 0.30 | 1.74 (1.12–2.70) | 1.69 (1.09–2.62) | 1.71 (1.10–2.65) | 1.66 (1.07–2.57) |
| **Acute myocardial infarction** | | | | | | |
| No PCOS | 3 502 | 0.21 | 1.0 (Reference) | 1.0 (Reference) | 1.0 (Reference) | 1.0 (Reference) |
| PCOS | 8 | 0.12 | 2.08 (1.04–4.17) | 2.10 (1.05–4.21) | 2.07 (1.03–4.15) | 2.07 (1.03–4.15) |
| **Stroke** | | | | | | |
| No PCOS | 13 105 | 0.80 | 1.0 (Reference) | 1.0 (Reference) | 1.0 (Reference) | 1.0 (Reference) |
| PCOS | 44 | 0.65 | 1.45 (1.08–1.95) | 1.26 (0.93–1.69) | 1.41 (1.05–1.90) | 1.24 (0.92–1.67) |
| **Hemorrhagic stroke** | | | | | | |
| No PCOS | 3 287 | 0.20 | 1.0 (Reference) | 1.0 (Reference) | 1.0 (Reference) | 1.0 (Reference) |
| PCOS | 13 | 0.19 | 1.42 (0.82–2.44) | 1.27 (0.74–2.19) | 1.38 (0.80–2.38) | 1.25 (0.72–2.16) |
| **Ischemic stroke** | | | | | | |
| No PCOS | 7 357 | 0.45 | 1.0 (Reference) | 1.0 (Reference) | 1.0 (Reference) | 1.0 (Reference) |
| PCOS | 29 | 0.43 | 1.88 (1.31–2.71) | 1.44 (1.00–2.08) | 1.82 (1.26–2.62) | 1.42 (0.98–2.05) |
| **Heart failure** | | | | | | |
| No PCOS | 4 827 | 0.30 | 1.0 (Reference) | 1.0 (Reference) | 1.0 (Reference) | 1.0 (Reference) |
| PCOS | 12 | 0.18 | 0.98 (0.56–1.73) | 0.87 (0.49–1.54) | 0.91 (0.52–1.61) | 0.84 (0.47–1.48) |
| **Atrial fibrillation** | | | | | | |
| No PCOS | 10 738 | 0.66 | 1.0 (Reference) | 1.0 (Reference) | 1.0 (Reference) | 1.0 (Reference) |
| PCOS | 18 | 0.27 | 1.05 (0.63–1.67) | 0.98 (0.62–1.56) | 1.02 (0.64–1.62) | 0.96 (0.60–1.52) |
| **Hypertensive disease** | | | | | | |
| No PCOS | 54 789 | 3.36 | 1.0 (Reference) | 1.0 (Reference) | 1.0 (Reference) | 1.0 (Reference) |
| PCOS | 146 | 2.16 | 1.75 (1.49–2.06) | 1.42 (1.20–1.67) | 1.63 (1.38–1.91) | 1.34 (1.14–1.58) |
| **Peripheral arterial disease** | | | | | | |
| No PCOS | 2 208 | 0.14 | 1.0 (Reference) | 1.0 (Reference) | 1.0 (Reference) | 1.0 (Reference) |
| PCOS | 6 | 0.09 | 1.18 (0.53–2.64) | 1.24 (0.56–2.77) | 1.17 (0.53–2.62) | 1.24 (0.55–2.77) |

Abbreviations: *PCOS* polycystic ovary syndrome, *CVD* cardiovascular disease, *HR* hazard ratio, *CI* confidence interval.
[a]Model 1 was unadjusted.
[b]Model 2 was adjusted for sex, country and calendar year of birth, maternal country of origin, parity, age, education, and marital status at the time of birth.
[c]Model 3 was adjusted for maternal hypertensive disorders, diabetes, and psychiatric disorders before or during the index pregnancy.
[d]Model 4 was adjusted for sex, country and calendar year of birth, maternal country of origin, parity, age, education, and marital status at the time of birth, hypertensive disorders, diabetes, and psychiatric disorders before or during the index pregnancy, and family history of cardiovascular disease.

population. Although the positive predictive value of the PCOS diagnosis in the national patient registers was reported to be high (86%)[29], it is likely that only women who sought medical care for clinically significant symptoms of PCOS or other medical conditions were diagnosed with PCOS; thus, the mothers with PCOS in our study may represent a less healthy group with more severe phenotypes. Women with milder PCOS symptoms who did not seek medical care or those who were diagnosed in private care might have been misclassified as not having PCOS. Similarly, only data from inpatient care and/or from the Swedish Medical Birth Register (MBR) were available for the first part of the study period. We expect this underreporting to result in an underestimation of the association between maternal PCOS and the risk of CVD in the offspring. Another reason for the lower prevalence of maternal PCOS in our study compared to that in a general population of women is related to the definition of our study population, i.e., a cohort of live births in two countries during almost five decades. Since the clinical manifestation of PCOS involves subfertility and infertility[30], it is expected that the prevalence of PCOS among mothers with live births is lower than that among women in the general population, on

which reports on the prevalence of PCOS are generally based. In addition, women without PCOS in our cohort were more likely to contribute with more births to the denominator of the PCOS prevalence rate than those with a PCOS diagnosis, potentially leading to a lower rate of maternal PCOS in this birth cohort than the rate of PCOS in a cohort of women from the total population. The prevalence of PCOS in our population is similar to that reported in previous Danish and Swedish register-based studies of a similar nature[6,31]. Furthermore, the diagnostic criteria for PCOS have changed over the study period. The Rotterdam criteria introduced in 2003 has a broader definition of PCOS than the National Institutes of Health criteria issued in 1990. However, the risk estimates for different PCOS diagnosis years (before 1990, 1990–2003, and after 2003) were comparable to each other and to the estimate for the overall period. Additionally, our sensitivity analyzes restricted to individuals born in Denmark after 1995 and in Sweden after 2001 yielded results consistent with those from the entire cohort.

Second, although the diagnoses of the major types of CVD (such as IHD and stroke) in the Danish and Swedish patient registers have

**Table 3 | Joint effect of maternal polycystic ovary syndrome and other comorbidities before childbirth on the risk of cardiovascular disease in offspring**

| Exposure | Number of events | Event rate, per 10000 person-years | HR (95% CI) | | | |
|---|---|---|---|---|---|---|
| | | | Model 1[a] | Model 2[b] | Model 3[c] | Model 4[d] |
| **Maternal PCOS and diabetes** | | | | | | |
| No PCOS, no diabetes | 378 834 | 23.92 | 1.0 (Reference) | 1.0 (Reference) | 1.0 (Reference) | 1.0 (Reference) |
| Only PCOS | 1 399 | 22.04 | 1.55 (1.48–1.64) | 1.21 (1.15–1.28) | 1.51 (1.44–1.59) | 1.20 (1.14–1.26) |
| Only diabetes | 3 948 | 24.72 | 1.41 (1.37–1.46) | 1.23 (1.19–1.27) | 1.37 (1.33–1.42) | 1.21 (1.18–1.25) |
| Both PCOS and diabetes | 93 | 29.79 | 2.63 (2.15–3.22) | 1.75 (1.43–2.14) | 2.43 (1.99–2.98) | 1.68 (1.37–2.06) |
| Synergy index and 95% CI for the additive interaction between PCOS and diabetes | | | 1.80 (1.75–1.85) | | | |
| **Maternal PCOS and hypertensive disease** | | | | | | |
| No PCOS, no hypertensive disease | 371 377 | 23.89 | 1.0 (Reference) | 1.0 (Reference) | 1.0 (Reference) | 1.0 (Reference) |
| Only PCOS | 1 369 | 22.16 | 1.58 (1.49–1.67) | 1.22 (1.16–1.29) | 1.54 (1.46–1.63) | 1.21 (1.15–1.28) |
| Only hypertensive disease | 11 405 | 25.12 | 1.33 (1.30–1.35) | 1.22 (1.19–1.24) | 1.31 (1.29–1.34) | 1.21 (1.19–1.23) |
| Both PCOS and hypertensive disease | 123 | 25.49 | 1.90 (1.59–2.27) | 1.44 (1.20–1.71) | 1.81 (1.51–2.16) | 1.40 (1.17–1.67) |
| Synergy index and 95% CI for the additive interaction between PCOS and hypertensive disease | | | 1.89 (1.75–2.04) | | | |
| **Maternal PCOS and overweight/obesity[e]** | | | | | | |
| No PCOS, no overweight/obesity | 77 063 | 20.30 | 1.0 (Reference) | 1.0 (Reference) | 1.0 (Reference) | 1.0 (Reference) |
| Only PCOS | 367 | 20.25 | 1.36 (1.23–1.51) | 1.16 (1.05–1.29) | 1.34 (1.21–1.48) | 1.15 (1.04–1.27) |
| Only overweight/obesity | 32 212 | 19.37 | 1.09 (1.08–1.11) | 1.02 (1.01–1.04) | 1.08 (1.06–1.09) | 1.01 (1.00–1.03) |
| Both PCOS and overweight/obesity | 530 | 21.14 | 1.48 (1.37–1.62) | 1.24 (1.14–1.35) | 1.42 (1.30–1.55) | 1.20 (1.10–1.30) |
| Synergy index and 95% CI for the additive interaction between PCOS and overweight/obesity | | | 2.23 (1.81–2.75) | | | |
| **Maternal PCOS and psychiatric disorders** | | | | | | |
| No PCOS, no psychiatric disorders | 372 232 | 23.93 | 1.0 (Reference) | 1.0 (Reference) | 1.0 (Reference) | 1.0 (Reference) |
| Only PCOS | 1 340 | 22.06 | 1.55 (1.47–1.64) | 1.22 (1.15–1.28) | 1.52 (1.44–1.60) | 1.20 (1.14–1.27) |
| Only psychiatric disorders | 10 551 | 23.92 | 1.41 (1.38–1.44) | 1.19 (1.17–1.22) | 1.40 (1.37–1.43) | 1.19 (1.17–1.21) |
| Both PCOS and psychiatric disorders | 152 | 25.98 | 2.32 (1.98–2.72) | 1.55 (1.32–1.81) | 2.24 (1.91–2.63) | 1.52 (1.29–1.78) |
| Synergy index and 95% CI for the additive interaction between PCOS and psychiatric disorders | | | 1.77 (1.69–1.86) | | | |

Abbreviations: *PCOS* polycystic ovary syndrome, *HR* hazard ratio, *CI* confidence interval.
[a]Model 1 was unadjusted.
[b]Model 2 was adjusted for sex, country and calendar year of birth, as well as maternal country of origin, parity, age, education, and marital status at the time of birth.
[c]Model 3 was adjusted for maternal hypertensive disease (except when examining its joint effect with polycystic ovary syndrome), diabetes (except when examining its joint effect with polycystic ovary syndrome), and psychiatric disorders (except when examining their joint effect with polycystic ovary syndrome) before or during the index pregnancy.
[d]Model 4 was adjusted for sex, country and calendar year of birth, as well as maternal country of origin, parity, age, education, and marital status at the time of birth, hypertensive disease (except when examining its joint effect with polycystic ovary syndrome), diabetes (except when examining its joint effect with polycystic ovary syndrome) and psychiatric disorders (except when examining their joint effect with polycystic ovary syndrome) before or during the index pregnancy, and family history of cardiovascular disease.
[e]This analysis was performed for individuals with data on maternal body-mass index available in early pregnancy (*N* = 3 274 570).

been reported to have high validity[32,33], a proportion of mild CVD cases are likely to have been missed due to unavailability of outpatient data in the first part of our study period. Nevertheless, when we restricted analyzes to the years when outpatient data were available, the associations did not change substantially. Similar to PCOS, there were changes over time in diagnostic criteria or diagnostic procedures of certain CVD types, such as AMI, hypertension, stroke, or heart failure. Therefore, we adjusted for calendar year at birth in our multivariable models to reduce the potential confounding by changing medical practices over time.

Third, we cannot rule out the possibility of uncontrolled confounding. Although we attempted to account for unmeasured confounding by familial characteristics using the cousin comparison design, this approach has limitations[34]. In general, cousin comparisons adjust for shared familial confounders less effectively than sibling comparisons, because cousin comparisons have limited ability to capture heterogeneity over time and within the

family unit. In addition, cousin comparisons are susceptible to bias from unmeasured confounders not shared by cousin pairs. Further, data on maternal smoking, BMI and the use of ART before the index birth were not available for all years of our study period, thus we could not control for these measures in our main analyzes. However, adjustment for maternal smoking, BMI or the use of ART in sensitivity analyzes among study participants with information on these variables did not substantially influence our estimates.

Fourth, although our sample size was substantial, the statistical power was limited in both the main analysis for certain CVD subtypes and in the cousin analysis. For example, the notably higher hazard ratios for IHD and AMI in the cousin analysis compared to those in the main analysis could potentially be chance findings. Future studies with greater statistical power, *e.g.*, when the cohorts included in the Nordic MBRs become older, may provide further insight into these questions.

**Table 4 | Hazard ratios and 95% confidence intervals for overall and specific cardiovascular disease, according to maternal polycystic ovary syndrome in the cousin analysis (N = 6 332 070)**

| Exposure | Number of events | Event rate, per 10 000 person-years | HR (95% CI) | | | |
|---|---|---|---|---|---|---|
| | | | Model 1[a] | Model 2[b] | Model 3[c] | Model 4[d] |
| **Overall CVD** | | | | | | |
| No PCOS | 352 298 | 23.69 | 1.0 (Reference) | 1.0 (Reference) | 1.0 (Reference) | 1.0 (Reference) |
| PCOS | 1 349 | 22.50 | 1.31 (1.18–1.45) | 1.19 (1.08–1.32) | 1.27 (1.16–1.42) | 1.18 (1.07–1.31) |
| **Ischemic heart disease** | | | | | | |
| No PCOS | 7 872 | 0.51 | 1.0 (Reference) | 1.0 (Reference) | 1.0 (Reference) | 1.0 (Reference) |
| PCOS | 20 | 0.31 | 4.22 (1.74–10.28) | 4.34 (1.77–10.63) | 4.22 (1.73–10.30) | 4.33 (1.76–10.65) |
| **Acute myocardial infarction** | | | | | | |
| No PCOS | 3 091 | 0.21 | 1.0 (Reference) | 1.0 (Reference) | 1.0 (Reference) | 1.0 (Reference) |
| PCOS | 8 | 0.13 | 10.37 (3.09–34.78) | 11.90 (3.49–40.55) | 10.56 (3.14–35.48) | 12.07 (3.54–41.22) |
| **Stroke** | | | | | | |
| No PCOS | 11 965 | 0.79 | 1.0 (Reference) | 1.0 (Reference) | 1.0 (Reference) | 1.0 (Reference) |
| PCOS | 38 | 0.62 | 0.63 (0.33–1.19) | 0.61 (0.32–1.16) | 0.64 (0.34–1.22) | 0.63 (0.33–1.19) |
| **Hemorrhagic stroke** | | | | | | |
| No PCOS | 3 005 | 0.20 | 1.0 (Reference) | 1.0 (Reference) | 1.0 (Reference) | 1.0 (Reference) |
| PCOS | 13 | 0.21 | 0.92 (0.32–2.62) | 0.87 (0.30–2.55) | 0.97 (0.34–2.78) | 0.93 (0.32–2.71) |
| **Ischemic stroke** | | | | | | |
| No PCOS | 6 703 | 0.44 | 1.0 (Reference) | 1.0 (Reference) | 1.0 (Reference) | 1.0 (Reference) |
| PCOS | 24 | 0.39 | 0.65 (0.28–1.49) | 0.61 (0.27–1.41) | 0.64 (0.28–1.48) | 0.61 (0.27–1.41) |
| **Heart failure** | | | | | | |
| No PCOS | 4 391 | 0.29 | 1.0 (Reference) | 1.0 (Reference) | 1.0 (Reference) | 1.0 (Reference) |
| PCOS | 10 | 0.16 | 0.61 (0.21–1.76) | 0.62 (0.21–1.77) | 0.61 (0.21–1.76) | 0.62 (0.21–1.78) |
| **Atrial fibrillation** | | | | | | |
| No PCOS | 9 530 | 0.63 | 1.0 (Reference) | 1.0 (Reference) | 1.0 (Reference) | 1.0 (Reference) |
| PCOS | 18 | 0.30 | 1.93 (0.91–4.12) | 1.99 (0.90–4.38) | 1.93 (0.91–4.09) | 1.99 (0.90–4.37) |
| **Hypertensive disease** | | | | | | |
| No PCOS | 49 308 | 3.26 | 1.0 (Reference) | 1.0 (Reference) | 1.0 (Reference) | 1.0 (Reference) |
| PCOS | 129 | 2.12 | 1.39 (0.98–1.96) | 1.30 (0.93–1.83) | 1.35 (0.95–1.91) | 1.27 (0.90–1.78) |
| **Peripheral arterial disease** | | | | | | |
| No PCOS | 2 013 | 0.13 | 1.0 (Reference) | 1.0 (Reference) | 1.0 (Reference) | 1.0 (Reference) |
| PCOS | 6 | 0.10 | 1.06 (0.30–3.77) | 1.13 (0.30–4.19) | 1.06 (0.30–3.78) | 1.13 (0.30–4.17) |

Abbreviations: *PCOS* polycystic ovary syndrome, *CVD* cardiovascular disease, *HR* hazard ratio; *CI* confidence interval.
[a]Model 1 was unadjusted.
[b]Model 2 was adjusted for sex, country and calendar year of birth, parity, age, education, and marital status at the time of birth.
[c]Model 3 was adjusted for maternal hypertensive disorders, diabetes, and psychiatric disorders before or during the index pregnancy.
[d]Model 4 was adjusted for sex, country and calendar year of birth, parity, age, education and marital status at the time of birth, hypertensive disorders, diabetes, and psychiatric disorders before or during the index pregnancy.

**Table 5 | Hazard ratios and 95% confidence intervals for overall cardiovascular disease according to maternal polycystic ovary syndrome mediated through preterm birth, small or large for gestational age, congenital heart disease, and diabetes**

| Mediators | HR (95% CI)[a] | | | Proportion mediated (%)[c] | P-value for the interaction between PCOS and the mediator[d] |
|---|---|---|---|---|---|
| | Total effect[b] | Direct effect | Mediated effect | | |
| Preterm birth | 1.19 (1.13–1.26) | 1.19 (1.13–1.25) | 1.01 (1.00–1.01) | 3.9 | 0.08 |
| SGA birth | 1.16 (1.10–1.23) | 1.16 (1.10–1.23) | 0.99 (0.99–1.00) | — | 0.14 |
| LGA birth | 1.19 (1.13–1.26) | 1.19 (1.12–1.26) | 1.01 (1.00–1.01) | 3.1 | 0.02 |
| Congenital heart disease | 1.19 (1.14–1.26) | 1.17 (1.11–1.24) | 1.02 (1.01–1.02) | 10.0 | 0.39 |
| Diabetes | 1.20 (1.14–1.26) | 1.19 (1.14–1.26) | 1.00 (0.99–1.00) | 0.78 | 0.51 |

Abbreviations: *PCOS* polycystic ovary syndrome, *SGA* small for gestational age, *LGA* large for gestational age, *HR* hazard ratio, *CI* confidence interval.
[a]We adjusted for sex, country and calendar year of birth, maternal country of origin, parity, age, education, and marital status at the time of birth, body-mass index and smoking during the index pregnancy, hypertensive disease, diabetes, and psychiatric disorders before or during the index pregnancy, and family history of cardiovascular diseases.
[b]The estimates for the total effect of maternal polycystic ovary syndrome on cardiovascular diseases may vary among models corresponding to different mediators because of the differences in the number of individual with missing data on each mediator and exposure-mediator interactions.
[c]The estimate of the proportion mediated by small for gestational age was outside the expected range of 0–100% since the direct effect and mediated effect were in opposite directions in this case.
[d]Wald tests were used to calculate two-sided *p*-values.

Finally, our findings may be directly generalizable only to children and young adults living in countries with sociocultural contexts and healthcare systems similar to those of Denmark and Sweden. Additionally, as there have been substantial changes in maternal age at childbirth and maternal BMI over the past five decades, the generalizability of the findings to children born in the current period is not clear.

In conclusion, we found that individuals born to mothers with PCOS had increased risks of CVD in childhood and young adulthood. The observed association could not be explained by familial confounding, PCOS-related comorbidities, or adverse pregnancy outcomes. If our findings are replicated and confirmed by future studies, children of women with PCOS may benefit from screening and early prevention for CVD.

## Methods

### Data sources and study population

Both Denmark and Sweden have a tax-funded healthcare system that provides free antenatal care to women[35]. The database used in the present study was obtained by linking data from several national registers in Denmark and Sweden using the unique personal identification number assigned to all residents of the two countries[36,37]. The registers used in the study are described in more detail in Supplementary Table 4. We included all live singleton births registered in the Danish MBR[38] and the Swedish MBR[39]) during 1973–2016 and 1973–2014, respectively. After excluding offspring of mothers with a missing or incomplete personal identification number, the final study population consisted of 6 839 703 live births. The study was reported to the Danish Data Protection Agency through registration at Aarhus University (record number: 2015-57-0002 · sequential number 654) and approved by the Research Ethics Committee at Karolinska Institute in Stockholm (No. 2016/288-31/1 and 2021-03315). The boards do not request informed consent for register-based studies.

### Measures

**Exposure.** Diagnoses of maternal PCOS were identified from the Danish National Patient Registry, the Swedish Patient Register, and the Swedish MBR, using the *International Statistical Classification of Diseases and Related Health Problems* (ICD) codes shown in Supplementary Table 5. We considered PCOS to be present during pregnancy regardless of the time of diagnosis, since PCOS is a disorder with metabolic disturbances and elevated testosterone levels that persist throughout the life of affected women[40].

**Outcomes.** We retrieved information on the first diagnosis of CVD from the national patient and cause-of-death registers, using the ICD codes listed in Supplementary Table 5. Further, we retrieved information on some important CVD types, *i.e.*, IHD (separately also for one of its main subtypes, i.e., AMI), stroke (separately also for ischemic stroke and hemorrhagic stroke), heart failure, hypertensive disease, atrial fibrillation (including atrial flutter), and peripheral artery disease. We followed study participants from birth until the first diagnosis of CVD, death, emigration (in Sweden data available until 2014), or the latest date with available data (December 31, 2016 in Denmark and December 31, 2020 in Sweden), whichever came first.

**Covariates.** Offspring's characteristics we retrieved information on were country of birth, year of birth, sex, birth weight, gestational age, and diagnoses of CHD and diabetes; maternal characteristics we obtained data on were country of origin, educational level, marital status, parity, age at the time of the index birth, BMI, smoking and ART during the index pregnancy, family history of CVD, and comorbidities including diabetes (including Type 1 and 2 diabetes), hypertensive disease, and psychiatric disorders before or during the index pregnancy. Preterm birth was defined as a gestational age <37 weeks at birth; SGA and LGA births were defined according to birth weight below or above the 10th percentile of the sex and gestational age-specific standard curve for normal fetal growth[41]. A detailed description of the measurement and categorization of covariates (including the ICD codes for the considered diseases) is provided in Supplementary note 1.

### Statistical analyzes

Incidence rates of CVD and its major subtypes were calculated as per 10 000 person-years. We performed Cox proportional hazard regression models to estimate HRs with 95% CIs for overall risks of CVD and its major subtypes, according to maternal PCOS, with attained age as the time scale. The proportional hazards assumption was examined using log-minus-log survival curves and Schoenfeld's residuals, and no violations were found. We ran several multivariate regression models: (1) Model 1 was unadjusted; (2) Model 2 was adjusted for offspring's sex, country and calendar year of birth, maternal country of origin, parity, age, education, and marital status at the time of birth; (3) Model 3 was adjusted for maternal hypertensive disorders, diabetes, and psychiatric disorders before or during the index pregnancy; and (4) Model 4 was adjusted for all covariates included in Models 2 and 3, and further adjusted for family history of CVD. Since information on maternal smoking and BMI during the index pregnancy was available only during part of the study period, we adjusted for these variables in sub-analyzes restricted to women with available data. We also split follow-up at 10, 20, 30, and 40 years to investigate whether the associations varied by attained age.

We examined the joint effects of maternal PCOS and other maternal comorbidities, *i.e.*, diabetes, hypertensive disease, overweight/obesity (BMI ≥ 25 kg/m$^2$) or psychiatric disorders on risk of CVD. Additive interactions between PCOS and comorbidities on CVD risk were examined using the synergy index; in case of this latter, departure from 1 suggests the presence of biological interaction[42].

To address potential confounding from shared familial factors, we conducted a cousin comparison analysis. First, we established a sub-cohort of cousin pairs, *i.e.*, the children of the mother and her biological sister(s), using personal identification numbers. Next, we ran stratified Cox models in this sub-cohort with a distinct stratum for each cousin pair, adjusting for the same covariates as in the overall population analysis, except for family history of CVD. Only cousin pairs discordant for exposure and outcome contributed to the estimates in the cousin comparison analyzes.

Maternal PCOS has been reported to be associated with preterm birth, SGA birth, LGA birth, CHD, and diabetes[6,10,43,44] which in turn are related to increased CVD risks later in offspring's lives[27,45,46]. Therefore, we performed a mediation analysis based on the counterfactual framework to explore the role of preterm birth, SGA or LGA birth, CHD, or diabetes before the CVD in the association between maternal PCOS and CVD in offspring. A detailed description of this approach is provided in Supplementary note 2.

We performed further stratified analyzes according to offspring's sex, or country of birth, or maternal use of ART and tested the interactions of these factors with maternal PCOS and CVD risk.

### Sensitivity analyzes

We created a PSM[47] sub-cohort to achieve a better balance of covariates between the exposed and unexposed groups and to further reduce potential confounding. We used logistic regression to estimate propensity scores as the probability of the mother having PCOS given certain covariates, *i.e.*, maternal country of origin, parity, education and marital status at birth, hypertensive disease, diabetes, and psychiatric disorders before or during the index pregnancy, and family history of CVD. We then used a ratio of 1:1 with a greedy matching

algorithm to match the exposed offspring and unexposed offspring, and finally created a propensity-score-matched sub-cohort. The characteristics of the PSM sub-cohort are described in detail in Supplementary Table 6. The Cox model was then used to estimate associations between maternal PCOS and risks of overall CVD and main CVD types in the PSM sub-cohort, with further adjustment for calendar year of birth, sex of the child, and parity.

In a further sensitivity analysis, we restricted to offspring who were born in Denmark since 1995 and in Sweden since 2001 when outpatient hospital data were available. To evaluate whether changes in diagnostic criteria influenced the estimates, we also categorized exposed offspring by year of PCOS diagnosis (*i.e.*, before 1990, 1990–2003, and after 2003, as diagnostic criteria for PCOS varied across these periods) and compared overall CVD risk across the three subgroups. As the validity of some of the CVD cases identified from the cause of death register may not be high, we conducted a sensitivity analysis, in which the outcome included only CVD cases identified from the national patient registers. Finally, we repeated our main analyzes after imputing missing data on covariates using the multivariate imputation by chained equations technique[48].

All data analyzes for this study were conducted using SAS, version 9.4 (SAS Institute Inc), and RStudio, version 1.2.1578 (RStudio Inc).

### Reporting summary
Further information on research design is available in the Nature Portfolio Reporting Summary linked to this article.

## Data availability
The raw Danish and Swedish cohort data were collected by national register holders in each country and cannot be shared publicly due to restrictions in our ethical approval and data privacy laws. The cohort data generated in this study have been securely stored on a secure server at Statistics Denmark (https://www.dst.dk/), which also prohibits data sharing with external users. Researchers who meet legal requirements may apply for similar data from the relevant registers, provided they have obtained the necessary ethical approvals. The holders of the registers used in this study and their websites are: Statistics Denmark (https://www.dst.dk/), Statistics Sweden (https://www.scb.se/en/), and the National Board of Health and Welfare in Sweden (https://www.socialstyrelsen.se/en/). Processing times for data applications to register holders, once ethical approvals are granted, vary depending on demand and the capacity of each register holder and may range from several months to over a year. The time periods for the availability of the granted data are regulated by agreements between register holders and the respective researchers. Statistics Denmark has strict access restrictions and Danish data cannot be exported or stored outside Statistics Denmark's server.

## Code availability
All data analyzes for this study were conducted on a secure server at Statistics Denmark. We are not allowed to export files from this server, neither data nor statistical codes.

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

## Acknowledgements

This study was supported by the Swedish Council for Working Life and Social Research (grant no. 2015-00837 to KDL), the Karolinska Institutet's Research Foundation (grant no. 2018-01924, 2018-01547 and 2022-01975 to KDL), the Swedish Heart and Lung Foundation (grant no. 20180306 and 20230493 to KDL), the China Scholarship Council (grant no. 201908300120 to FY and grant no. 202008310043 to ZLW), the Novo Nordisk Foundation (grant no. NNF18OC0052029 to JL), the Independent Research Fund Denmark (grant no. DFF 9039-00010B and 1030-00012B to JL), Nordic Cancer Union (R275-A15770 and R278-A15877 to JL), and the National Key Research and Development Program of China (grant no. 2021YFC2701003 to MHM). The funders had no role in the design and conduct of the study; collection, management, analysis, and interpretation of the data; preparation, review, or approval of the manuscript; or decision to submit the manuscript for publication.

## Author contributions

F.Y. had full access to all the data in the study and takes responsibility for the integrity of the data and the accuracy of the data analysis. I.J., J.L., and K.D.L. conceived and designed the study. F.Y. performed the data management and the statistical analyzes. M.G., J.L., and K.D.L. provided administrative, technical, and material support. I.J., J.L., and K.D.L. supervised the study. F.Y., Z.L.W., H.T.S., I.J., J.L., and K.D.L. contributed to analyzes or interpretation of the results. F.Y., Z.L.W., and K.D.L. drafted the manuscript. F.Y., Z.L.W., H.T.S., I.J., M.G., W.Y., M.H.M., N.R., A.K.W., J.L., and K.D.L. critically revised it for important intellectual content.

## Funding

## Competing interests

The authors declare no competing interests.
