## [Transparent Peer Review file · Nature Communications]

Maternal polycystic ovary syndrome and offspring's risk of cardiovascular diseases in childhood and young adulthood

Corresponding Author: Dr Fen Yang

Version 0:

Reviewer comments:

Reviewer #1

(Remarks to the Author)

This manuscript reported the results of a data analysis examining the relationship between maternal polycystic ovary syndrome (PCOS) and offspring's risk of cardiovascular diseases in childhood and young adulthood. Utilizing national register-based cohort data from Denmark and Sweden, the strengths of this study include a large sample size with long follow-up from childhood to early middle age, the availability of many important confounders, and a number of thoughtful sensitivity analysis.

In general, this manuscript is well written and clearly presented. The findings of a positive association between maternal PCOS and CVD risk in their offsprings are relatively novel and have the potential to significantly advance the current literature in relevant research fields. However, my enthusiasm for this manuscript is reduced by a few major concerns:

1. The most important concern I have is the validity of major statistical analysis conducted. Although Cox regression model with age as the time scale is a proper statistical model to use for the purpose of this study, some of the analysis results presented seem to be questionable. For example, in Table 2, the estimated rates per 10,000 person years of the outcomes are usually lower in the PCOS group than the no PCOS group. This is surprising as one would expect to see higher incidence rates of CVD among the offsprings with maternal PCOS than those without maternal PCOS. Meanwhile, most of the unadjusted hazard ratios are estimated to be greater than 1. Again, this is inconsistent with the event rates reported in the same table, as lower event rates in the PCOS group usually means the hazard ratio of the outcome would be less than 1 for PCOS. In other words, based on the numbers reported in the event rate column, I would expect PCOS was associated with lower risk of CVD in the unadjusted models. But this was not case based on the unadjusted hazard ratios. I think the authors need to double check if they conducted the statistical analysis properly and/or reported the number of events or event rates correctly.
2. Only 2 models were presented in Table 2-4. It'll be helpful to present a few more adjusted models: such as those only adjusted by demographics and those only adjusted by comorbidities etc... It'll also be helpful to further discuss the potential confounding effects of different covariates included these models.
3. The descriptions of the cousin comparison analysis and propensity score matched analysis were too brief and need to be expanded to provide more details regarding those sensitivity analyses.
4. Table 4: The estimated hazard ratios from the cousin analysis were not completely similar to those in the main analysis. For example, the hazard ratios for ischemic heart disease were much higher in the cousin analysis than those in the main analysis. Furthermore, the hazard ratios for stroke in the cousin analysis were lower than 1, but they are higher than 1 in the main analysis even though the event rates were pretty similar in both analyses. Also, why does Table 4 include fewer CVD subtypes than Table 2?
5. It'll be helpful to add another sensitivity analysis, stratifying the analysis by country.
6. The clinical implications for the synergistic effects of maternal PCOS and other comorbidities should be more thoroughly discussed in the Discussion section.

Reviewer #2

(Remarks to the Author)

This is a binational registry study on CVD outcomes in offspring of women with PCOS. It is an important topic in a unique long term integrated data set.

It advances knowledge, however there are significant limitations that need to be acknowledged.

The major challenges include:

the captured PCOS prevalence is 16 fold lower than is documented internationally, reporting on a very highly selected population and leaving 15/16 cases in the "non PCOS " group - this is unavoidable in registry data, and whilst it still makes a contribution to the literature, the paper does not adequately acknowledge / discuss this limitation

Please clarify how congenital heart disease was managed (these offspring should be excluded)

Over interpretation of analyses based on very small case numbers with some over stated interpretation in the text

Additional points:

Recommend noting current International Guideline criteria endorsed by 39 international Societies since 2018 and reinforced in 2023. Indeed the manuscript does not mention the International guidelines endorsed by the Nordic Federation of OG, which includes significantly updated references compared to those used here and current systematic reviews including CVD risk in those with PCOS and their relatives. This data advances knowledge in this area and should be grounded on this recent background of recent systematic reviews.

The introduction focuses on in-utero environment, but neglects considerable cardiometabolic genetic risk of offspring. Please add a section on this (such as Zhu et al JCEM 2022). This cannot all be accounted for by cousin analysis, especially given the small numbers.

Methods

Table 2, 3 and 4 - what about maternal BMI? This is important to consider or capture as a limitation clearly.

Numbers are very small in tables 2.3 and 4 bringing into question the validity of this analysis. Please justify this, it is suggested that some sub analyses are removed and this should be noted as a very considerable limitation.

Results

The marked underestimate of PCOS prevalence (12%) and here sitting at 0.76% undervalues the significance of these results. This needs to be noted in the abstract.

Please add clarification for the reader on why the event rates are lower for 10,000 person years yet the HR are higher in PCOS offspring even in the unadjusted model.

The cousin analysis methods, results and implications are not clear. Please add detail to enable interpretation, for example results in cousin pairs were quoted as similar to associations in the main analysis - overall the HR was similar, but sub analyses results HRs vary from 0.65 to 4.11, with very small numbers. In the text it is reported throughout that cousin analysis was "similar to the main results" and did not support a genetic origin of CVD in offspring. I would suggest this analysis again has small sample sizes and the strength of conclusions is too strong.

Sex differences here are limited - this is an important finding and could be expanded in the discussion including links back to polygenic risk scores (as noted above). Indeed with a male polygenic risk score now identified, this is important to explore further.

Limitations need to be modified. The prevalence of PCOS is 16 fold lower than global prevalence estimates (see 2023 International guidelines). Very low event numbers need to be emphasised. Maternal age and BMI differ considerably to contemporary times (most mothers here were <30 years and had BMIs < 8% obese in the non PCOS cohort and 25% in the PCOS group. Age of offspring remains a major limitation many of the references are outdated and have been superseded including systematic reviews - especially aligned to the 2023 guideline

Please replace reference 6

Most of the references focus on Nordic data and whilst understandable in this context, it limits the generalisability of the discussion

Reviewer #3

(Remarks to the Author)

This is a well written and very thoroughly analyzed manuscript on an interesting topic. The tables and results are very clearly presented. Their cohort is the great fit for this analysis. Although it is very thorough, I just have a few more comments that could add to the analysis or discussion:

1. The decision to include psychiatric diagnoses as a cofounder was not discussed and was not included in the DAG. Why was this included.
2. Was there any ability to include a diagnosis of PCOS or diabetes in the offspring? I can understand why logistically this probably doesn't work well, but you may want to discuss it if not show the percentages, if possible.
3. During the time period of this study, there were significant changes in the ability to diagnose small acute MI's with the implementation of the use of troponin. You discuss the changes over time in the diagnosis of PCOS but not of this. Possibly you could add this to the discussion.
4. It is mentioned that cause of death registries were used to define CVD. This can be problematic as CVD is often used as a cause of death when it is unexpected or unknown. The details of this process and how this was incorporated or how many of the CVD outcomes were defined this way.
5. You show an unadjusted analysis and a fully adjusted analysis. Presumably, the unadjusted is confounded so reviewing the adjusted analysis, it shows that PAD, and stroke were not significant, so I would not say there is an association.

Version 1:

Reviewer comments:

Reviewer #1

(Remarks to the Author)

This revision has addressed most of my previous concerns. However, I have a major concern that was not completely addressed. This concern is regarding the discrepancy between unadjusted hazard ration and the event rate of CVD presented in Table 2. I agree this discrepancy is likely related to confounding by age. And in fact, I just noticed that the prevalence of maternal PCOS increase quickly with calendar year of birth, as shown in Table 1. Specifically, the prevalence of maternal PCOS among those born in 2009-2016 was >40 times of that among those born in 1973-1978. As explained by the authors, the diagnostic criteria for PCOS have changed substantially over the study period and that's likely the major reason for the substantial changes in PCOS prevalence over time. However, given this fact, I don't think combining all the years of data is a proper way of analyzing this data. The severe underdiagnosis of PCOS in earlier years could substantially bias the estimates for the associations of interest. I would strongly recommend the authors to only focus on data after 2003 to ensure the validity of the conclusions of this study. The earlier years of data could be included in sensitivity analyses, but should not be used for the main analyses and major conclusions.

Also, is it unclear what's the stratification factor used eTable 4. What does "attained age" mean? The ages of attaining CVD or PCOS? And either way, what's the age used for those who did not have CVD or maternal PCOS? A more appropriate stratified analysis could be a cox regression model stratified by the calendar year of birth and should be included as a sensitivity analysis.

Reviewer #2

(Remarks to the Author)

The authors responses and modifications have improved the paper, however for reviewer 2 a few areas need further revision.

#1- There are no references for the claims around higher nulliparity and lower fecundity. The literature shows that those with PCOS have similar nulliparity rates, albeit with OI or ART, than those without PCOS. At most, the literature shows a 4% difference. Family size is also a little lower but largely similar. These reasons are inadequate and do not explain the poor capture rate of PCOS status. The NIH diagnosis point is valid., whereby a prevalence of around 8% vs 12% on Rotterdam, would see the proportion of cases captured in controls of at around 1 case for every 10 captured in the controls. This is ubiquitous in studies of this nature but specifically needs to be stated in terms of its impact here. The magnitude of the case load in the controls needs to be specified and recognised.

Comment 2- CHD is almost always structural and is a totally different condition to metabolic CVD, IHD, hypertensive HD and stroke.. Including CHD because it is related to the same organ as metabolic CVD and may be on the "same causal pathway" does not appear to be a clinically valid argument. Ideally the CHD would be excluded as the rationale provided does not make clinical sense.

#6 BMI is inexorably linked to CVD and to PCOS. a comment in the limitations on the lack of BMI data is needed. This includes the low rate of obesity compared to current prevalence.

#8 as noted above number of children per women in PCOS will have marginal impact here and should be removed or deemphasised in the rationale for very low capture rates. The authors may wish to highlight that their work does align to similar prevalence capture in large population studies of this nature.

Reviewer #3

(Remarks to the Author)

I approve of the revisions. My only issue is that I think that the term "trend toward an association" for results that are not significant is valid. Although often said, I think trend should be reserved for changes over time, or explained as to what you are referring to as a trend.

Version 2:

Reviewer comments:

Reviewer #1

(Remarks to the Author)

Although this revision has addressed some of my previous concerns, my major concern was not completely addressed. In particular, I don't understand why the authors could not conduct a sensitivity analysis by restricting the study sample to those whose birth year was after 2003. Although that'll result in a smaller sample size with shorter follow-up years, it'll still be a sample of about 2 million individuals. And even if the parameter estimate has a wider confidence interval due to much lower power, the magnitude of the estimated hazard ratio in this sensitivity analysis should be similar to the main analysis if the conclusion of this study is robust. Or another approach is to conduct stratified analysis, stratifying by year of birth (before 1990, 1990-2003, and after 2003) to see if the hazard ratios in different strata are relatively consistent across different calendar years of birth.

Reviewer #4

(Remarks to the Author)

With regard to comment 2 re CHD, I would agree with the original reviewer that CHD is not on the causal pathway for CVD and I would have thought that those with CHD should be excluded. The mechanisms causing CVD in those with CHD would surely be different to those without CHD. However, the authors have substantiated their views to justify why those with CHD should not be excluded.

I believe the authors have satisfactorily addressed comments to reviewer 2.

Version 3:

Reviewer comments:

Reviewer #1

(Remarks to the Author)

This revision has adequately addressed my previous concerns. I have no additional comments except that I suggest the authors moving Table 1 in the response letter to the supplementary tables. Also, they should add a few sentences discussing the results of sensitivity analysis (Table 1 in the response letter) in the Discussion section.

Response to the reviewers' comments

Manuscript's ID: NCOMMS-23-60298A

Title: Maternal polycystic ovary syndrome and offspring's risk of cardiovascular diseases in childhood and young adulthood

Reviewer #1 (Remarks to the Author):

Comment:

1. This manuscript reported the results of a data analysis examining the relationship between maternal polycystic ovary syndrome (PCOS) and offspring's risk of cardiovascular diseases in childhood and young adulthood. Utilizing national register-based cohort data from Denmark and Sweden, the strengths of this study include a large sample size with long follow-up from childhood to early middle age, the availability of many important confounders, and a number of thoughtful sensitivity analysis.

In general, this manuscript is well written and clearly presented. The findings of a positive association between maternal PCOS and CVD risk in their offsprings are relatively novel and have the potential to significantly advance the current literature in relevant research fields.

Response:

Thank you for these positive and constructive comments.

Comment:

2. However, my enthusiasm for this manuscript is reduced by a few major concerns:

The most important concern I have is the validity of major statistical analysis conducted. Although Cox regression model with age as the time scale is a proper statistical model to use

for the purpose of this study, some of the analysis results presented seem to be questionable. For example, in Table 2, the estimated rates per 10,000 person years of the outcomes are usually lower in the PCOS group than the no PCOS group. This is surprising as one would expect to see higher incidence rates of CVD among the offsprings with maternal PCOS than those without maternal PCOS. Meanwhile, most of the unadjusted hazard ratios are estimated to be greater than 1. Again, this is inconsistent with the event rates reported in the same table, as lower event rates in the PCOS group usually means the hazard ratio of the outcome would be less than 1 for PCOS. In other words, based on the numbers reported in the event rate column, I would expect PCOS was associated with lower risk of CVD in the unadjusted models. But this was not case based on the unadjusted hazard ratios. I think the authors need to double check if they conducted the statistical analysis properly and/or reported the number of events or event rates correctly.

Response:

We acknowledge the discrepancy between the reported incidence rates in the two exposure groups and the corresponding hazard ratios in Table 2. We agree that this initially may appear counterintuitive. We have again thoroughly reviewed our statistical procedures and SAS code and believe that the reported event rates and the hazard ratios are correct.

Several factors influence the incidence rates of cardiovascular disease (CVD) in the two exposure groups and the associated hazard ratios. Most importantly, confounding by age is likely to be an important explanation for the discrepancy between the incidence rates and the “crude” hazard ratios given that (1) age has a well-established strong association with the risk of CVD, but also (2) there are likely to be differences over time in the diagnosis and the registration of PCOS in this cohort. As seen in Table 1, the birth year distribution differed substantially between the exposure groups, resulting in corresponding differences in their age distribution. The higher proportion of younger individuals in the exposed than in the unexposed group is likely to have contributed to the lower overall incidence rate of CVD in the exposed group. In contrast, Cox regression, when using age as the underlying time scale, inherently adjusts for age. Therefore, even the “crude” hazard ratio in our Cox model accounted for confounding by age.

To further address concerns regarding potential confounding by age, we conducted stratified analyses according to finer age groups of offspring (i.e. <10 years, 10-19 years, 20-29 years,

30-39 years, and ≥ 40 years). This additional analysis revealed consistently higher cumulative incidence rates of CVD in the exposed than the unexposed group across all age strata, and corresponding hazard ratios above 1 (Supplementary eTable 4). These findings corroborate our explanation for the discrepancy between the observed lower overall incidence rate of CVD in the exposed than in the unexposed group, compared to the corresponding hazard ratios being above 1 (Table 2).

Comment:

3. Only 2 models were presented in Table 2-4. It'll be helpful to present a few more adjusted models: such as those only adjusted by demographics and those only adjusted by comorbidities etc... It'll also be helpful to further discuss the potential confounding effects of different covariates included these models.

Response:

Based on the Reviewer's suggestion, we now include two additional models in Tables 2-4, *i.e.*, Model 2 in which only demographic characteristics were adjusted for and Model 3 in which only maternal diseases were adjusted for. Tables 2-4 now present results for Model 1 (unadjusted), Model 2 (adjusted for sex, country and calendar year of birth, maternal country of origin, parity, age, education, and marital status at the time of birth), Model 3 (adjusted for maternal hypertensive disorders, diabetes disorders, and psychiatric disorders before or during the index pregnancy), and Model 4 (adjusted for all covariates included in Models 2 and 3, and further adjusted for family history of cardiovascular disease). Hazard ratios in Models 1-4 were generally comparable, though in some sub-analyses, the association was somewhat weaker in models adjusted for demographic factors.

We have now revised the text of the manuscript as follows:

- In the Methods section: "We ran several multivariate regression models: (1) Model 1 was unadjusted; (2) Model 2 was adjusted for offspring's sex, country and calendar year of birth, maternal country of origin, parity, age, education, and marital status at the time of birth; (3) Model 3 was adjusted for maternal hypertensive disorders, diabetes, and psychiatric disorders before or during the index pregnancy; and (4) Model 4 was adjusted

for all covariates included in Models 2 and 3, and further adjusted for family history of CVD.” (Page 7, Lines 16-21).

- In the Results section: “Adjustment for potential confounders did not substantially change the estimates.” (Page 11, Lines 2-3).
- The results are presented in Tables 2-4:

Table 2. Hazard ratios and 95% confidence intervals for overall and specific cardiovascular diseases in offspring according to maternal polycystic ovary syndrome

Exposure	Number of events	Event rate, per 10 000 person-years	HR (95% CI)			
			Model 1 ^a	Model 2 ^b	Model 3 ^c	Model 4 ^d
Overall CVD						
No PCOS	382 782	23.93	1.0 (Reference)	1.0 (Reference)	1.0 (Reference)	1.0 (Reference)
PCOS	1492	22.41	1.59 (1.51-1.67)	1.23 (1.17-1.29)	1.53 (1.45-1.61)	1.21 (1.15-1.27)
Ischemic heart disease						
No PCOS	8640	0.53	1.0 (Reference)	1.0 (Reference)	1.0 (Reference)	1.0 (Reference)
PCOS	20	0.30	1.74 (1.12-2.70)	1.69 (1.09-2.62)	1.71 (1.10-2.65)	1.66 (1.07-2.57)
Acute myocardial infarction						
No PCOS	3502	0.21	1.0 (Reference)	1.0 (Reference)	1.0 (Reference)	1.0 (Reference)
PCOS	8	0.12	2.08 (1.04-4.17)	2.10 (1.05-4.21)	2.07 (1.03-4.15)	2.07 (1.03-4.15)
Stroke						
No PCOS	13 105	0.80	1.0 (Reference)	1.0 (Reference)	1.0 (Reference)	1.0 (Reference)
PCOS	44	0.65	1.45 (1.08-1.95)	1.26 (0.93-1.69)	1.41 (1.05-1.90)	1.24 (0.92-1.67)
Hemorrhagic stroke						
No PCOS	3287	0.20	1.0 (Reference)	1.0 (Reference)	1.0 (Reference)	1.0 (Reference)

PCOS	13	0.19	1.42 (0.82-2.44)	1.27 (0.74-2.19)	1.38 (0.80-2.38)	1.25 (0.72-2.16)
Ischemic stroke						
No PCOS	7357	0.45	1.0 (Reference)	1.0 (Reference)	1.0 (Reference)	1.0 (Reference)
PCOS	29	0.43	1.88 (1.31-2.71)	1.44 (1.00-2.08)	1.82 (1.26-2.62)	1.42 (0.98-2.05)
Heart failure						
No PCOS	4827	0.30	1.0 (Reference)	1.0 (Reference)	1.0 (Reference)	1.0 (Reference)
PCOS	12	0.18	0.98 (0.56-1.73)	0.87 (0.49-1.54)	0.91 (0.52-1.61)	0.84 (0.47-1.48)
Atrial fibrillation						
No PCOS	10 738	0.66	1.0 (Reference)	1.0 (Reference)	1.0 (Reference)	1.0 (Reference)
PCOS	18	0.27	1.05 (0.63-1.67)	0.98 (0.62-1.56)	1.02 (0.64-1.62)	0.96 (0.60-1.52)
Hypertensive disease						
No PCOS	54 789	3.36	1.0 (Reference)	1.0 (Reference)	1.0 (Reference)	1.0 (Reference)
PCOS	146	2.16	1.75 (1.49-2.06)	1.42 (1.20-1.67)	1.63 (1.38-1.91)	1.34 (1.14-1.58)
Peripheral arterial disease						
No PCOS	2208	0.14	1.0 (Reference)	1.0 (Reference)	1.0 (Reference)	1.0 (Reference)
PCOS	6	0.09	1.18 (0.53-2.64)	1.24 (0.56-2.77)	1.17 (0.53-2.62)	1.24 (0.55-2.77)

Abbreviations: PCOS, polycystic ovary syndrome; CVD, cardiovascular disease; HR, hazard ratio; CI, confidence interval.

^a Model 1 was unadjusted.

^b Model 2 was adjusted for sex, country and calendar year of birth, maternal country of origin, parity, age, education, and marital status at the time of birth.

^c Model 3 was adjusted for maternal hypertensive disorders, diabetes, and psychiatric disorders before or during the index pregnancy.

^d Model 4 was adjusted for sex, country and calendar year of birth, maternal country of origin, parity, age, education, and marital status at the time of birth, hypertensive disorders, diabetes, and psychiatric disorders before or during the index pregnancy, and family history of cardiovascular disease.

Table 3. Joint effect of maternal polycystic ovary syndrome and other comorbidities before childbirth on the risk of cardiovascular disease in offspring

Exposure	Number of events	Event rate, per 10 000 person-years	HR (95% CI)				
			Model 1 ^a	Model 2 ^b	Model 3 ^c	Model 4 ^d	
Maternal PCOS and diabetes							
No PCOS, no diabetes	378 834	23.92	1.0 (Reference)	1.0 (Reference)	1.0 (Reference)	1.0 (Reference)	
Only PCOS	1 399	22.04	1.55 (1.48-1.64)	1.21 (1.15-1.28)	1.51 (1.44-1.59)	1.20 (1.14-1.26)	
Only diabetes	3 948	24.72	1.41 (1.37-1.46)	1.23 (1.19-1.27)	1.37 (1.33-1.42)	1.21 (1.18-1.25)	
Both PCOS and diabetes	93	29.79	2.63 (2.15-3.22)	1.75 (1.43-2.14)	2.43 (1.99-2.98)	1.68 (1.37-2.06)	
Synergy index and 95% CI for the additive interaction between PCOS and diabetes				1.80 (1.75-1.85)			
Maternal PCOS and hypertensive disease							
No PCOS, no hypertensive disease	371 377	23.89	1.0 (Reference)	1.0 (Reference)	1.0 (Reference)	1.0 (Reference)	
Only PCOS	1 369	22.16	1.58 (1.49-1.67)	1.22 (1.16-1.29)	1.54 (1.46-1.63)	1.21 (1.15-1.28)	
Only hypertensive disease	11 405	25.12	1.33 (1.30-1.35)	1.22 (1.19-1.24)	1.31 (1.29-1.34)	1.21 (1.19-1.23)	
Both PCOS and hypertensive disease	123	25.49	1.90 (1.59-2.27)	1.44 (1.20-1.71)	1.81 (1.51-2.16)	1.40 (1.17-1.67)	
Synergy index and 95% CI for the additive interaction between PCOS and hypertensive disease				1.89 (1.75-2.04)			
Maternal PCOS and overweight/obesity ^e							
No PCOS, no overweight/obesity	77 063	20.30	1.0 (Reference)	1.0 (Reference)	1.0 (Reference)	1.0 (Reference)	
Only PCOS	367	20.25	1.36 (1.23-1.51)	1.16 (1.05-1.29)	1.34 (1.21-1.48)	1.15 (1.04-1.27)	
Only overweight/obesity	32 212	19.37	1.09 (1.08-1.11)	1.02 (1.01-1.04)	1.08 (1.06-1.09)	1.01 (1.00-1.03)	
Both PCOS and overweight/obesity	530	21.14	1.48 (1.37-1.62)	1.24 (1.14-1.35)	1.42 (1.30-1.55)	1.20 (1.10-1.30)	
Synergy index and 95% CI for the additive interaction between PCOS and overweight/obesity				2.23 (1.81-2.75)			
Maternal PCOS and psychiatric disorders							

No PCOS, no psychiatric disorders	372 232	23.93	1.0 (Reference)	1.0 (Reference)	1.0 (Reference)	1.0 (Reference)
Only PCOS	1 340	22.06	1.55 (1.47-1.64)	1.22 (1.15-1.28)	1.52 (1.44-1.60)	1.20 (1.14-1.27)
Only psychiatric disorders	10 551	23.92	1.41 (1.38-1.44)	1.19 (1.17-1.22)	1.40 (1.37-1.43)	1.19 (1.17-1.21)
Both PCOS and psychiatric disorders	152	25.98	2.32 (1.98-2.72)	1.55 (1.32-1.81)	2.24 (1.91-2.63)	1.52 (1.29-1.78)

Synergy index and 95% CI for the additive

interaction between PCOS and psychiatric disorders

1.77 (1.69-1.86)

Abbreviations: PCOS, polycystic ovary syndrome; HR, hazard ratio; CI, confidence interval.

^a Model 1 was unadjusted.

^b Model 2 was adjusted for sex, country and calendar year of birth, as well as maternal country of origin, parity, age, education, and marital status at the time of birth.

^c Model 3 was adjusted for maternal hypertensive disease (except when examining its joint effect with polycystic ovary syndrome), diabetes (except when examining its joint effect with polycystic ovary syndrome), and psychiatric disorders (except when examining their joint effect with polycystic ovary syndrome) before or during the index pregnancy.

^d Model 4 was adjusted for sex, country and calendar year of birth, as well as maternal country of origin, parity, age, education, and marital status at the time of birth, hypertensive disease (except when examining its joint effect with polycystic ovary syndrome), diabetes (except when examining its joint effect with polycystic ovary syndrome) and psychiatric disorders (except when examining their joint effect with polycystic ovary syndrome) before or during the index pregnancy, and family history of cardiovascular disease.

^e This analysis was performed for individuals with maternal body-mass index data available in early pregnancy (N=3 274 570).

Table 4. Hazard ratios and 95% confidence intervals for overall and specific cardiovascular disease, according to maternal polycystic ovary syndrome in the cousin analysis (N=6 332 070)

Exposure	Number of events	Event rate, per 10 000 person-years	HR (95% CI)			
			Model 1 ^a	Model 2 ^b	Model 3 ^c	Model 4 ^d
Overall CVD						
No PCOS	352 298	23.69	1.0 (Reference)	1.0 (Reference)	1.0 (Reference)	1.0 (Reference)

PCOS	1349	22.50	1.31 (1.18-1.45)	1.19 (1.08-1.32)	1.27 (1.16-1.42)	1.18 (1.07-1.31)
Ischemic heart disease						
No PCOS	7872	0.51	1.0 (Reference)	1.0 (Reference)	1.0 (Reference)	1.0 (Reference)
PCOS	20	0.31	4.22 (1.74-10.28)	4.34 (1.77-10.63)	4.22 (1.73-10.30)	4.33 (1.76-10.65)
Acute myocardial infarction						
No PCOS	3091	0.21	1.0 (Reference)	1.0 (Reference)	1.0 (Reference)	1.0 (Reference)
PCOS	8	0.13	10.37 (3.09-34.78)	11.90 (3.49-40.55)	10.56 (3.14-35.48)	12.07 (3.54-41.22)
Stroke						
No PCOS	11 965	0.79	1.0 (Reference)	1.0 (Reference)	1.0 (Reference)	1.0 (Reference)
PCOS	38	0.62	0.63 (0.33-1.19)	0.61 (0.32-1.16)	0.64 (0.34-1.22)	0.63 (0.33-1.19)
Hemorrhagic stroke						
No PCOS	3005	0.20	1.0 (Reference)	1.0 (Reference)	1.0 (Reference)	1.0 (Reference)
PCOS	13	0.21	0.92 (0.32-2.62)	0.87 (0.30-2.55)	0.97 (0.34-2.78)	0.93 (0.32-2.71)
Ischemic stroke						
No PCOS	6703	0.44	1.0 (Reference)	1.0 (Reference)	1.0 (Reference)	1.0 (Reference)
PCOS	24	0.39	0.65 (0.28-1.49)	0.61 (0.27-1.41)	0.64 (0.28-1.48)	0.61 (0.27-1.41)
Heart failure						
No PCOS	4391	0.29	1.0 (Reference)	1.0 (Reference)	1.0 (Reference)	1.0 (Reference)
PCOS	10	0.16	0.61 (0.21-1.76)	0.62 (0.21-1.77)	0.61 (0.21-1.76)	0.62 (0.21-1.78)
Atrial fibrillation						
No PCOS	9530	0.63	1.0 (Reference)	1.0 (Reference)	1.0 (Reference)	1.0 (Reference)
PCOS	18	0.30	1.93 (0.91-4.12)	1.99 (0.90-4.38)	1.93 (0.91-4.09)	1.99 (0.90-4.37)
Hypertensive disease						

No PCOS	49 308	3.26	1.0 (Reference)	1.0 (Reference)	1.0 (Reference)	1.0 (Reference)
PCOS	129	2.12	1.39 (0.98- 1.96)	1.30 (0.93- 1.83)	1.35 (0.95- 1.91)	1.27 (0.90- 1.78)
Peripheral arterial disease						
No PCOS	2013	0.13	1.0 (Reference)	1.0 (Reference)	1.0 (Reference)	1.0 (Reference)
PCOS	6	0.10	1.06 (0.30- 3.77)	1.13 (0.30- 4.19)	1.06 (0.30- 3.78)	1.13 (0.30- 4.17)

Abbreviations: PCOS, polycystic ovary syndrome; CVD, cardiovascular disease; HR, hazard ratio; CI, confidence interval.

^a Model 1 was unadjusted.

^b Model 2 was adjusted for sex, country and calendar year of birth, parity, age, education, and marital status at the time of birth.

^c Model 3 was adjusted for maternal hypertensive disorders, diabetes, and psychiatric disorders before or during the index pregnancy.

^d Model 4 was adjusted for sex, country and calendar year of birth, parity, age, education and marital status at the time of birth, hypertensive disorders, diabetes, and psychiatric disorders before or during the index pregnancy.

Comment:

4. The descriptions of the cousin comparison analysis and propensity score matched analysis were too brief and need to be expanded to provide more details regarding those sensitivity analyses.

Response:

Based on the Reviewer's suggestion, we have expanded the description of the cousin comparison analysis in the Methods section, as follows: "To address potential confounding from shared familial factors, we conducted a cousin comparison analysis. First, we established a sub-cohort of cousin pairs, *i.e.*, the children of the mother and her biological sister(s), using personal identification numbers. Next, we ran stratified Cox models in this sub-cohort with a distinct stratum for each cousin pair, adjusting for the same covariates as in the overall population analysis, except for family history of CVD. Only cousin pairs discordant for exposure and outcome contributed to the estimates in the cousin comparison analyses." (Page 8, Lines 10-16).

A detailed description of the propensity-score matched analysis now is provided in the Methods section: “We used logistic regression to estimate propensity scores as the probability of the mother having PCOS given certain covariates, *i.e.*, maternal country of origin, parity, education and marital status at birth, hypertensive disease, diabetes, and psychiatric disorders before or during the index pregnancy, and family history of CVD. We then used a ratio of 1:1 with a greedy matching algorithm to match the exposed offspring and unexposed offspring, and finally created a propensity-score-matched sub-cohort.” (Page 9, Lines 10-15).

Comment:

5. Table 4: The estimated hazard ratios from the cousin analysis were not completely similar to those in the main analysis. For example, the hazard ratios for ischemic heart disease were much higher in the cousin analysis than those in the main analysis. Furthermore, the hazard ratios for stroke in the cousin analysis were lower than 1, but they are higher than 1 in the main analysis even though the event rates were pretty similar in both analyses. Also, why does Table 4 include fewer CVD subtypes than Table 2?

Response:

We have now added all CVD subtypes to Tables 2-4. The updated Table 4 was presented in our response to the third comment from this Reviewer. We agree that the estimated hazard ratios from the cousin analysis were not identical to those in the main analysis; we have therefore revised the text as follows:

“In the cousin analysis, most associations were attenuated, but the association of maternal PCOS with the risk of overall CVD, IHD, AMI, and hypertensive disease persisted. The hazard ratios for IHD and AMI in the cousin analysis were greater than the corresponding hazard ratios in the main analysis (Table 4).” (Page 11, Lines 19-20; Page 12, Lines 1-2).

Comment:

6. It'll be helpful to add another sensitivity analysis, stratifying the analysis by country.

Response:

We now have performed a sensitivity analysis according to offspring's country of birth. The results are presented in Supplementary eTable 5:

eTable 5. Sub-analyses of the association between maternal polycystic ovary syndrome and cardiovascular disease in offspring (Selected part of the whole table)

Exposure	Number of events	Event rate, per 10 000 person-years	HR (95% CI)	
			Model 1 ^a	Model 2 ^b
Stratified analysis according to offspring's country of birth				
Denmark				
No PCOS	130 768	23.36	1.0 (Reference)	1.0 (Reference)
PCOS	451	20.88	1.50 (1.37-1.65)	1.20 (1.09-1.31)
Sweden				
No PCOS	252 014	24.22	1.0 (Reference)	1.0 (Reference)
PCOS	1041	23.14	1.55 (1.46-1.64)	1.21 (1.13-1.28)
P-value for multiplicative interaction between PCOS and country of birth			0.01	0.03

Abbreviations: PCOS, polycystic ovary syndrome; ART, assisted reproductive treatment; HR, hazard ratio; CI, confidence interval.

^a Model 1 was unadjusted.

^b Model 2 was adjusted for sex, country and calendar year of birth, maternal origin of country, parity, age, education, and marital status at the time of birth, hypertensive disorders, diabetes, and psychiatric disorders before or during the index pregnancy, and family history of cardiovascular diseases.

Comment:

7. The clinical implications for the synergistic effects of maternal PCOS and other comorbidities should be more thoroughly discussed in the Discussion section.

Response:

We have revised the Discussion section to address the clinical implications of the synergistic effects of maternal PCOS and comorbidities: “However, when we investigated the joint effect of PCOS and comorbidities, we observed synergism, *i.e.*, the risks of CVD were higher among offspring of women with both PCOS and a comorbidity than would be expected from the effects of each exposure alone. In other words, the adverse effects of these comorbidities on cardiovascular risk were more pronounced among women with PCOS than among women without PCOS. Further research is necessary to elucidate the mechanisms driving this synergism. Nevertheless, these findings may highlight the importance of proactive clinical management, particularly for pregnant women with multiple complications. Early screening and preventive measures among children born to mothers with PCOS and comorbidities may have the potential to mitigate their long-term cardiovascular consequences.” (Page 14, Lines 10-19).

Reviewer #2 (Remarks to the Author):**Comment:**

This is a binational registry study on CVD outcomes in offspring of women with PCOS. It is an important topic in a unique long term integrated data set. It advances knowledge, however there are significant limitations that need to be acknowledged.

The major challenges include:

1. The captured PCOS prevalence is 16 fold lower than is documented internationally, reporting on a very highly selected population and leaving 15/16 cases in the “non PCOS” group – this is unavoidable in registry data, and whilst it still makes a contribution to the literature, the paper does not adequately acknowledge / discuss this limitation.

Response:

We acknowledge the difference in the observed prevalence of maternal PCOS in our study population compared to broader international figures in women. As suggested by the Reviewer, this discrepancy may arise in part due to potential underrepresentation of mild

cases in our registers or those managed solely in primary care settings. Additionally, given our study question, women in our study had to have at least one live birth. As PCOS is linked with infertility and subfertility, a substantial proportion of women with PCOS, especially those with a severe condition, do not become pregnant and would not be included in a study such as ours. In our cohort, women without PCOS could have had a higher number of births and contribute with several births to the denominator of the prevalence rate of maternal PCOS, potentially resulting in reduced prevalence of maternal PCOS.

Furthermore, changes in diagnostic criteria for PCOS (from those issued by the National Institutes of Health criteria in 1990 to the Rotterdam criteria issued in 2003) further contributed to the low number of captured cases over the study period.

We have addressed these concerns in the Limitations section: “The study had several limitations. First, the prevalence of maternal PCOS in our cohort is lower than that reported in the general population. Although the positive predictive value of the PCOS diagnosis in the national patient registers was reported to be high (86%), it is likely that only women who sought medical care for clinically significant symptoms of PCOS or other medical conditions were diagnosed with PCOS; thus, the mothers with PCOS in our study may represent a less healthy group with more severe phenotypes. Another reason for the lower prevalence of maternal PCOS in our study compared to that in a general population of women is related to the definition of our study population, i.e. a cohort of live births in two countries during almost five decades. Since the clinical manifestation of PCOS involves subfertility and infertility, it is expected that the prevalence of PCOS among mothers with live births is much lower than that among women in the general population, on which reports on the prevalence of PCOS are generally based. Furthermore, women who contributed with multiple births to our cohort were less likely to have PCOS than women excluded due to lack of a live birth or women with only one child. Fertile and healthier women tend to have more births, potentially leading to a lower proportion of maternal PCOS in our birth cohort than the PCOS prevalence in the general population of women. Furthermore, the diagnostic criteria for PCOS have changed over the study period. The Rotterdam criteria introduced in 2003 has a broader definition of PCOS than the National Institutes of Health criteria issued in 1990. However, the risk estimates for different PCOS diagnosis years (before 1990, 1990-2003, and after 2003) were comparable to each other and to the estimate for the overall period.” (Page 16, Lines 16-21; Page 17, Lines 1-14).

Comment:

2. Please clarify how congenital heart disease was managed (these offspring should be excluded)

Response:

We addressed the role of the offspring's congenital heart disease (CHD) as described in the Methods section: "Maternal PCOS has been reported to be associated with preterm birth, SGA birth, LGA birth, CHD, and diabetes, which in turn are related to increased CVD risks later in offspring's lives. Therefore, we performed a mediation analysis based on the counterfactual framework to explore the role of preterm birth, SGA or LGA birth, CHD, or diabetes before the CVD in the association between maternal PCOS and CVD in offspring." (Page 8, Lines 17-20).

It is important to note that we did not exclude offspring with CHD from our analysis, but rather hypothesized that this variable may be on the causal pathway between maternal PCOS and CVD risk in offspring. Our primary objective in this study was to estimate the total effect of maternal PCOS on risk of CVD in offspring; thus, excluding these cases would have resulted in estimating only the direct effect of maternal PCOS (without mediation through CHD) on offspring's CVD risk. We therefore treated CHD as a potential mediator and conducted mediation analysis to examine its contribution to the association between maternal PCOS and CVD risk in offspring.

Comment:

3. Over interpretation of analyses based on very small case numbers with some over stated interpretation in the text.

Response:

We now have revised the text to avoid overinterpretation of results from sub-analyses with small sample sizes:

- In the Results section: “Compared with unexposed offspring, those exposed to maternal PCOS had higher risks of overall CVD (HR in fully adjusted model, 1.21, 95% CI, 1.14 to 1.27) and of some CVD subtypes, including ischemic heart disease, acute myocardial infarction, and hypertensive disease; there was a trend towards an association in the case of stroke (Table 2).” (Page 11, Lines 3-6).
- In the Results section: “In the cousin analysis, most associations were attenuated, but the association of maternal PCOS with the risk of overall CVD, IHD, AMI, and hypertensive disease persisted. The hazard ratios for IHD and AMI in the cousin analysis were stronger than corresponding hazard ratios in the main analysis (Table 4).” (Page 11, Lines 19-20; Page 12, Lines 1-2).
- In the Discussion section: “The comparable hazard ratios for overall CVD and several CVD types, including IHD, AMI, and hypertensive disease, from the main analysis and the cousin comparison analysis suggest that shared familial factors may not fully account for the association between maternal PCOS and the risk of CVD in offspring.” (Page 13, Lines 6-9).
- In the Discussion section: “Fourth, although our sample size was substantial, the statistical power was limited in both the main analysis for certain CVD subtypes and in the cousin analysis. For example, the notably higher hazard ratios for IHD and AMI in the cousin analysis compared to the main analysis could potentially be chance findings. Future studies with greater statistical power, *e.g.*, when the cohorts included in the Nordic Medical Birth Registers become older, may provide further insight into these questions.” (Page 18, Lines 13-18).

Comment:

4. Additional points:

Recommend noting current International guidelines criteria endorsed by 39 international Societies since 2018 and reinforced in 2023. Indeed, the manuscript does not mention the International guidelines endorsed by the Nordic Federation of OG, which includes significantly updated references compared to those used here and current systematic reviews including CVD risk in those with PCOS and their relatives. This data advances knowledge in this area and should be grounded on this recent background of recent systematic reviews.

Response:

We appreciate the Reviewer's suggestion that we incorporate the latest guidelines for PCOS, including those endorsed by the Nordic Federation of Obstetrics and Gynecology, in our manuscript. We have carefully reviewed these guidelines and have updated the Introduction section, including prevalence and clinical features of PCOS, as follows: "Polycystic ovary syndrome (PCOS) is one of the most common endocrine disorders in women of reproductive age, and its estimated prevalence ranges from 10% to 13%. According to the latest International Evidence-based guidelines for the Assessment and Management of PCOS, PCOS is a heterogeneous condition characterized by several reproductive, cardiometabolic, and psychological disorders, including infertility, obesity, diabetes, cardiovascular disease (CVD), depression, and anxiety." (Page 4, Lines 2-7).

Comment:

5. The introduction focuses on in-utero environment, but neglects considerable cardiometabolic genetic risk of offspring. Please add a section on this (such as Zhu et al JCEM 2022). This cannot all be accounted for by cousin analysis, especially given the small numbers.

Response:

We value the Reviewer's input and the suggested reference, which we thoroughly reviewed. The referenced study indicates that genetic traits linked to PCOS, as reflected by polygenic risk scores, may influence shared metabolic dysfunctions in both male and female offspring. Therefore, together with the in-utero effects of PCOS, genetic predisposition also may play a significant role in cardiometabolic disease development.

As our primary objective was to examine the long-term impact of the in-utero environment on cardiovascular health, we considered genetic predisposition as a potential confounding factor. To address this concern, we conducted a cousin analysis to partially mitigate the confounding effects of genetic predisposition. To clarify our rationale for employing cousin analysis, we have revised the Introduction section. We now include the reference suggested by the Reviewer and add the following statement: "In addition, as there is a familial predisposition to

cardiometabolic diseases in offspring born to women with PCOS, we used a cousin comparison design to assess whether familial genetic or environmental characteristics contribute to the association.” (Page 5, Lines 3-5). We extended our discussion of the role of confounding factors as follows: “First, shared genetic predisposition or environmental risk factors, such as maternal lifestyle and socioeconomic status, may contribute to the observed association between maternal PCOS and CVD risk in offspring. However, we observed no substantial changes in the associations after adjusting for several maternal characteristics or performing cousin-comparison analyses. Nevertheless, the role of genetic confounding cannot be excluded in the present study.” (Page 15, Lines 7-12).

Comment:

6. Methods

Table 2, 3 and 4 - what about maternal BMI? This is important to consider or capture as a limitation clearly.

Response:

Given the availability of data on maternal body-mass index (BMI) only since 2003 in Denmark and since 1982 in Sweden, we chose not to include this variable in our main analyses due to power considerations. However, to address concerns about residual confounding by maternal BMI, we conducted a sensitivity analysis in which we adjusted for maternal BMI in study participants with available data. We note in the Methods section that “Since information on maternal smoking and BMI during the index pregnancy was available only during part of the study period, we adjusted for these variables in sub-analyses restricted to women with available data.” (Page 8, Lines 1-3).

The results of these analyses, presented in Supplementary eTable 5, indicated that the results did not substantially change after adjusting for maternal BMI in addition to the factors included in the main model, suggesting that BMI may not strongly confound the associations. Therefore, lack of adjustment for BMI in the main analysis is unlikely to significantly alter the results.

Furthermore, in Table 3, which presents the results of the joint effect of maternal PCOS and other comorbidities before childbirth on the risk of CVD in offspring, we specifically considered maternal BMI $\geq 25\text{kg/m}^2$ as indicative of overweight/obesity. We explored its joint effect with maternal PCOS on offspring's CVD risk.

Comment:

7. Numbers are very small in tables 2.3 and 4 bringing into question the validity of this analysis. Please justify this, it is suggested that some sub analyses are removed and this should be noted as a very considerable limitation.

Response:

For the main outcome, *i.e.* overall CVD, we believe that our sample size was generally adequate to perform both the main and sub-analyses. However, we acknowledge that for certain CVD subtypes, the limited number of cases potentially could have influenced the validity of our results. Because of the associated considerations about statistical power, we chose to perform sub-analyses (excluding the cousin analysis) only for overall CVD. We now acknowledge this limitation in the Discussion section as follows: “Fourth, although our sample size was substantial, the statistical power was limited in both the main analysis for certain CVD subtypes and in the cousin analysis. For example, the notably higher hazard ratios for IHD and AMI in the cousin analysis compared to the main analysis could potentially be chance findings. Future studies with greater statistical power, *e.g.*, when the cohorts included in the Nordic Medical Birth Registers become older, may provide further insight into these questions.” (Page 18, Lines 13-18).

Comment:

8. Results

The marked underestimate of PCOS prevalence (12%) and here sitting at 0.76% undervalues the significance of these results. This needs to be noted in the abstract.

Response:

In the Results section of the Abstract, we note that the prevalence of maternal PCOS in our study population was 0.8%. In the Discussion section of the text, we then provide a detailed discussion of potential reasons for the lower prevalence of maternal PCOS in this cohort of live births compared to the general population of women. For more information, please refer to our response to this Reviewer's first comment.

Comment:

9. Please add clarification for the reader on why the event rates are lower for 10,000 person years yet the HR are higher in PCOS offspring even in the unadjusted model.

Response:

Please refer to our response to the first Reviewer's second comment.

Comment:

10. The cousin analysis methods, results and implications are not clear. Please add detail to enable interpretation, for example results in cousin pairs were quoted as similar to associations in the main analysis - overall the HR was similar, but sub analyses results HRs vary from 0.65 to 4.11, with very small numbers. In the text it is reported throughout that cousin analysis was "similar to the main results" and did not support a genetic origin of CVD in offspring. I would suggest this analysis again has small sample sizes and the strength of conclusions is too strong.

Response:

We have now revised the corresponding text as follows:

- In the Results section: "In the cousin analysis, most associations were attenuated, but the association of maternal PCOS with the risk of overall CVD, IHD, AMI, and hypertensive

disease persisted. The hazard ratios for IHD and AMI in the cousin analysis were stronger than the corresponding hazard ratios in the main analysis (Table 4).” (Page 11, Lines 19-20; Page 12, Lines 1-2).

- In the Discussion section: “The comparable hazard ratios for overall CVD and several CVD types, including IHD, AMI, and hypertensive disease, from the main analysis and the cousin comparison analysis suggest that shared familial factors may not fully account for the association between maternal PCOS and the risk of CVD in offspring.” (Page 13, Lines 6-9).

Comment:

11. Sex differences here are limited - this is an important finding and could be expanded in the discussion including links back to polygenic risk scores (as noted above). Indeed with a male polygenic risk score now identified, this is important to explore further.

Response:

We have now extended our discussion regarding sex differences as:

“Our findings of no sex differences in the association between maternal PCOS and CVD risk in offspring is consistent with results of several earlier studies suggesting that the effects of polygenic risk of PCOS on cardiometabolic health were similar in men and women. However, other studies suggest the potential for a sex-specific effect; some found that males have a greater cardiovascular risk compared to females at the same level of genetic predisposition for PCOS, while others have observed altered cardiometabolic features predominantly among girls born to women with PCOS. Further research is needed to elucidate these sex differences.” (Page 14, Lines 20-21; Page 15, Lines 1-6).

Comment:

12. Limitations need to be modified. The prevalence of PCOS is 16 fold lower than global prevalence estimates (see 2023 International guideline). Very low event numbers need to be

emphasised. Maternal age and BMI differ considerably to contemporary times (most mothers here were <30 years and had BMIs < 8% obese in the non PCOS cohort and 25% in the PCOS group. Age of offspring remains a major limitation

Response:

We have discussed the limitation about the low prevalence of maternal PCOS in our cohort, low number of events in case of some CVD subtypes, young age of offspring, and the changing maternal age and BMI in the Discussion section:

- “First, the prevalence of maternal PCOS in our cohort is lower than that reported in the general population. Although the positive predictive value of the PCOS diagnosis in the national patient registers was reported to be high (86%), it is likely that only women who sought medical care for clinically significant symptoms of PCOS or other medical conditions were diagnosed with PCOS; thus, the selected mothers with PCOS may represent a less healthy group with more severe phenotypes. Another reason for the lower prevalence of maternal PCOS in our study compared to that in a general population of women is related to the definition of our study population, i.e. a cohort of live births in two countries during almost five decades. Since the clinical manifestation of PCOS involves subfertility and infertility, it is expected that the prevalence of PCOS among mothers with live births is much lower than that among women in the general population, on which reports on the prevalence of PCOS are generally based. Furthermore, women who contributed with multiple births to our cohort were less likely to have PCOS than women excluded due to lack of a live birth or women with only one child. Fertile and healthier women tend to have more births, potentially leading to a lower proportion of maternal PCOS in our birth cohort than the PCOS prevalence in the general population of women. Furthermore, the diagnostic criteria for PCOS have changed over the study period. The Rotterdam criteria introduced in 2003 has a broader definition of PCOS than the National Institutes of Health criteria issued in 1990. However, the risk estimates for different PCOS diagnosis years (before 1990, 1990-2003, and after 2003) were comparable among each other and to the estimate for the overall period.” (Page 16, Lines 16-21; Page 17, Lines 1-14).
- “Fourth, although our sample size was substantial, the statistical power was limited in both the main analysis for certain CVD subtypes and in the cousin analysis. For example, the notably higher hazard ratios for IHD and AMI in the cousin analysis compared to the main

analysis could potentially be chance findings. Future studies with greater statistical power, e.g. when the cohorts involved in the Nordic Medical Birth Registers become older, may provide further insights into these questions.” (Page 18, Lines 13-18).

- “Finally, our findings may be directly generalizable only to children and young adults living in countries with sociocultural contexts and healthcare systems similar to those of Denmark and Sweden. Additionally, it is important to note that maternal age at childbirth and maternal BMI in our study cohort may vary from current norms.” (Page 18, Lines 19-21).

Comment:

13. many of the references are outdated and have been superseded including systematic reviews - especially aligned to the 2023 guidelines

Response:

We have now omitted some outdated references and replaced them with more current and relevant ones.

Comment:

14. Please replace reference 6

Response:

We have now replaced the previous reference with a later publication: *Gunning MN, et al. Cardiometabolic health in offspring of women with PCOS compared to healthy controls: a systematic review and individual participant data meta-analysis. Hum Reprod Update. 2020 Jan 1;26(1):103-117.*

Comment:

15. Most of the references focus on Nordic data and whilst understandable in this context, it limits the generalizability of the discussion

Response:

We now have added relevant studies from non-Nordic populations to enhance the generalizability of our Discussion.

Reviewer #3 (Remarks to the Author):

Comment:

This is a well written and very thoroughly analyzed manuscript on an interesting topic. The tables and results are very clearly presented. Their cohort is the great fit for this analysis. Although it is very thorough, I just have a few more comments that could add to the analysis or discussion:

1. The decision to include psychiatric diagnoses as a cofounder was not discussed and was not included in the DAG. Why was this included.

Response:

In the Directed acyclic graph (DAG), which was based on previous research, we drew an arrow from maternal psychiatric disorders (referred as “m_dis”, including maternal hypertensive disorders, diabetes, and psychiatric disorders) to offspring’s CVD to reflect an association between maternal psychiatric disorders and risk of CVD in their children. However, we did not include a direct link between maternal PCOS and psychiatric disorders in the DAG, as it remains uncertain whether PCOS causes psychiatric disorders or vice versa. What we do know is that women with PCOS are more prone to comorbidities such as obesity, diabetes, and psychiatric disorders. Thus, it is unclear whether maternal psychiatric disorders could act as confounders or effect modifier or both in the observed association between maternal PCOS and CVD risk. We adjusted for maternal psychiatric disorders in the multivariable model to account for its effect. We also performed stratified and interaction analyses to investigate whether maternal psychiatric disorders could modify the association between maternal PCOS and CVD in offspring.

Comment:

2. Was there any ability to include a diagnosis of PCOS or diabetes in the offspring? I can understand why logistically this probably doesn't work well, but you may want to discuss it if not show the percentages, if possible.

Response:

Given the increased risk of PCOS in offspring born to mothers with PCOS (*Boldis BV, et al. Early Life Factors and Polycystic Ovary Syndrome in a Swedish Birth Cohort. Int J Environ Res Public Health. 2023 Nov 20;20(22):708*), and the reported association between PCOS and CVD (*Ollila MM, et al. Women with PCOS have an increased risk for cardiovascular disease regardless of diagnostic criteria-a prospective population-based cohort study. Eur J Endocrinol. 2023 Jul 20;189(1):96-105*), the offspring's PCOS status could serve as a potential mediator. This could be explored through mediation analysis. However, in our study a significant portion of the offspring were too young to receive a PCOS diagnosis, as PCOS symptoms typically manifest after puberty. Therefore, identifying offspring with PCOS solely through patient registers may result in severe misclassification. Moreover, some offspring without a PCOS diagnosis during our study period may receive a diagnosis in the future, which would not be captured if it falls outside our observation period. Thus, performing the mediation analysis for the role of offspring's PCOS may underestimate its mediated effect.

We now address this in the Discussion section: "Furthermore, daughters of women with PCOS are at an increased risk of developing PCOS, which, in turn, could be associated with subsequent CVD risk. Unfortunately, we could not investigate the role of offspring's PCOS in the association between maternal PCOS and offspring's CVD. The reason is that a high proportion of the female offspring were too young to receive a PCOS diagnosis, as symptoms typically emerge post-puberty. Additionally, some offspring without a PCOS diagnosis during our study period may receive a diagnosis in the future, beyond our observation period and the period of data availability. The potential misclassification of PCOS among offspring could lead to underestimation of its mediated effect." (Page 15, Lines 12-20).

We agree that diabetes in offspring also could act as a mediator since offspring of women with PCOS are at increased risk of this condition (*Chen X, et al. Association of maternal polycystic ovary syndrome or anovulatory infertility with obesity and diabetes in offspring: a population-based cohort study. Hum Reprod. 2021 Jul 19;36(8):2345-2357*). Diabetes in turn is a well-known risk factor for CVD. To follow up on the Reviewer’s comment we now have conducted an additional mediation analysis to assess the contribution of diabetes in offspring to the observed association between maternal PCOS and CVD risk in offspring. The results indicate that the contribution of diabetes may be modest, with the proportion mediated by diabetes estimated at 0.78%. We have now revised the text in the manuscript accordingly:

- In the Methods section: “Offspring’s characteristics we retrieved information on were country of birth, year of birth, sex, birth weight, gestational age, and diagnoses of congenital heart disease (CHD) and diabetes; ...” (Page 6, Lines 16-18).
- In the Methods section: “Maternal PCOS has been reported to be associated with preterm birth, SGA births, LGA births, CHD, and diabetes, which in turn are related to increased CVD risks later in offspring’s lives. Therefore, we performed a mediation analysis based on the counterfactual framework to explore the role of preterm birth, SGA or LGA birth, CHD, or diabetes before the CVD in the association between maternal PCOS and CVD in offspring.” (Page 8, Lines 17-20).
- In the Results section: “The association between maternal PCOS and risk of CVD was largely independent of preterm birth, SGA or LGA birth, CHD, or diabetes. Congenital heart disease mediated 10.0% of the observed association, but there was limited evidence for mediation in the case of the other four conditions (Table 5).” (Page 12, Lines 5-8).
- In the Discussion section: “Similarly, we found no evidence of mediation by diabetes.” (Page 16, Line 7).

Table 5. Hazard ratios and 95% confidence intervals for overall cardiovascular disease according to maternal polycystic ovary syndrome mediated through preterm birth, small or large for gestational age, congenital heart disease, and diabetes

Mediators	HR (95% CI) a			Proportion mediated (%) c	P-value for the interaction between PCOS and the mediator
	Total effect b	Direct effect	Mediated effect		
Preterm birth	1.19 (1.13-1.26)	1.19 (1.13-1.25)	1.01 (1.00-1.01)	3.9	0.08
SGA birth	1.16 (1.10-1.23)	1.16 (1.10-1.23)	0.99 (0.99-1.00)	-	0.14
LGA birth	1.19 (1.13-1.26)	1.19 (1.12-1.26)	1.01 (1.00-1.01)	3.1	0.02

Congenital heart disease	1.19 (1.14-1.26)	1.17 (1.11-1.24)	1.02 (1.01-1.02)	10.0	0.39
Diabetes	1.20 (1.14-1.26)	1.19 (1.14-1.26)	1.00 (0.99-1.00)	0.78	0.51

Abbreviations: PCOS, polycystic ovary syndrome; SGA, small for gestational age; LGA, large for gestational age; HR, hazard ratio; CI, confidence interval.

^a We adjusted for sex, country and calendar year of birth, maternal country of origin, parity, age, education, and marital status at the time of birth, body-mass index and smoking during the index pregnancy, hypertensive disease, diabetes, and psychiatric disorders before or during the index pregnancy, and family history of cardiovascular diseases.

^b The estimates for the total effect of maternal polycystic ovary syndrome on cardiovascular diseases may vary among models corresponding to different mediators because of the differences in the number of individual with missing data on each mediator and exposure-mediator interactions.

^c The estimate of the proportion mediated by small for gestational age was outside the expected range of 0-100% since the direct effect and mediated effect were in opposite directions in this case.

Comment:

3. During the time period of this study, there were significant changes in the ability to diagnose small acute MI's with the implementation of the use of troponin. You discuss the changes over time in the diagnosis of PCOS but not of this. Possibly you could also add this to the discussion.

Response:

We acknowledge the Reviewer’s concern regarding the potential influence of temporal changes in medical practices on the relationship between maternal PCOS and offspring’s CVD risk. To address this concern, we incorporated calendar year at birth as a covariate in our multivariable models. We have updated the manuscript with a discussion of this concern: “Similar to PCOS, there were changes over time in diagnostic criteria or diagnostic procedures of certain CVD types, such as AMI, hypertension, stroke, or heart failure. Therefore, we adjusted for calendar year at birth in our multivariable models to reduce the potential confounding by changing medical practices over time.” (Page 17, Lines 20-22).

Comment:

4. It is mentioned that cause of death registries were used to define CVD. This can be problematic as CVD is often used as a cause of death when it is unexpected or unknown. The details of this process and how this was incorporated or how many of the CVD outcomes were defined this way.

Response:

Among the 384,274 CVD cases identified in our study, 1244 cases (3.2%) were based on data from the cause of death registers.

To address concerns associated with the validity of the CVD cases identified from the cause of death register, we have now conducted a sensitivity analysis, that included only CVD cases identified from the national patient register. The results of this sensitivity analysis were similar to those of the main analysis, which included CVD cases from both the patient register and the cause of death register. We have revised the text accordingly:

- In the Methods section: “As the validity of some of the CVD cases identified from the cause of death register may not be high, we conducted a sensitivity analysis, in which the outcome included only CVD cases identified from the national patient registers.” (Page 10, Lines 4-6).
- In the Results section: “The results did not substantially change when we... (4) restricted our definition of CVD to cases identified from the national patient registers (n=383,030); ...” (Page 12, Lines 17-18).

eTable 5. Sub-analyses of the association between maternal polycystic ovary syndrome and cardiovascular disease in offspring

Exposure	Number of events	Event rate, per 10 000 person-years	HR (95% CI)	
			Model 1 ^a	Model 2 ^b
Restricted to CVD cases identified from national patient registers				
No PCOS	381 542	23.32	1.0 (Reference)	1.0 (Reference)
PCOS	1488	21.39	1.54 (1.46-1.62)	1.20 (1.14-1.27)

Abbreviations: PCOS, polycystic ovary syndrome; ART, assisted reproductive treatment; HR, hazard ratio; CI, confidence interval.

^aModel 1 was unadjusted.

^bModel 2 was adjusted for sex, country and calendar year of birth, maternal origin of country, parity, age, education, and marital status at the time of birth, hypertensive disorders, diabetes disorders, and psychiatric disorders before or during the index pregnancy, and family history of cardiovascular diseases.

Comment:

5. You show an unadjusted analysis and a fully adjusted analysis. Presumably, the unadjusted is confounded so reviewing the adjusted analysis, it shows that PAD, and stroke were not significant, so I would not say there is an association.

Response:

We have now revised the text in the Results section accordingly: “Compared with unexposed offspring, those exposed to maternal PCOS had higher risks of overall CVD (HR in fully adjusted model, 1.21, 95% CI, 1.14 to 1.27) and of some CVD subtypes, including ischemic heart disease, acute myocardial infarction, and hypertensive disease; there was a trend towards an association in case of stroke (Table 2).” (Page 11, Lines 3-6).

REVIEWER COMMENTS

Reviewer #1 (Remarks to the Author):

Comment:

1. This revision has addressed most of my previous concerns. However, I have a major concern that was not completely addressed. This concern is regarding the discrepancy between unadjusted hazard ratios and the event rate of CVD presented in Table 2. I agree this discrepancy is likely related to confounding by age. And in fact, I just noticed that the prevalence of maternal PCOS increases quickly with calendar year of birth, as shown in Table 1. Specifically, the prevalence of maternal PCOS among those born in 2009-2016 was >40 times of that among those born in 1973-1978. As explained by the authors, the diagnostic criteria for PCOS have changed substantially over the study period and that's likely the major reason for the substantial changes in PCOS prevalence over time. However, given this fact, I don't think combining all the years of data is a proper way of analyzing this data. The severe underdiagnosis of PCOS in earlier years could substantially bias the estimates for the associations of interest. I would strongly recommend the authors to only focus on data after 2003 to ensure the validity of the conclusions of this study. The earlier years of data could be included in sensitivity analyses, but should not be used for the main analyses and major conclusions.

Response:

We appreciate the Reviewer's continued attention to the discrepancy between the cardiovascular disease (CVD) incidence rates in the two exposure groups and the corresponding hazard ratios in Table 2. We have explained, as acknowledged by the Reviewer, that this discrepancy primarily arises from the difference in age distribution between the groups exposed to maternal polycystic ovary syndrome (PCOS), and those not exposed. Due to the change in diagnostic criteria for PCOS over the study period, and probably also due to the improved sensitivity of the PCOS diagnoses in the registers in later years (with the availability of specialised outpatient clinic data), there was a higher prevalence of younger individuals (born in later years) in the exposed than in the unexposed group. Consequently, the incidence rates were influenced not only by exposure status but also by age distribution of the exposed and the unexposed groups. We therefore advise against fully relying on these

incidence rates. Our primary objective in this study was to examine the hazard ratios for CVD by exposure to maternal PCOS. The Cox model that we applied, using age as the underlying time scale, accounted for potential age-related confounding.

To mitigate the impact of changing diagnostic criteria and differences in the accuracy of the register-based data over time, we employed several strategies:

- 1) We included calendar year of birth as a covariate in our main multivariable models. This adjustment allowed to compute the CVD risk associated with exposure to maternal PCOS while holding constant other covariates, including calendar year of birth.
- 2) We performed an analysis according to the year of PCOS diagnosis (i.e., before 1990, 1990-2003, and after 2003) to take into account differences in diagnostic criteria. The risk estimates for these subgroups were comparable to each other and to the overall estimate.
- 3) We performed analyses restricted to those born since 1995 in Denmark and since 2001 in Sweden, i.e. to the periods when the patient registers of the two countries also included diagnoses from specialised outpatient clinic care.

We have now extended the discussion in the Limitations section about the very relevant concern that the underdiagnosis of PCOS in earlier periods may have resulted in an underestimation of the association between maternal PCOS and CVD risk in the offspring: “... Similarly, only data from inpatient care and/or from the Swedish MBR were available for the first part of the study period. We expect this underreporting to result in an underestimation of the association between maternal PCOS and the risk of CVD in the offspring.” (Please see page 17, line 8-11).

While we appreciate the suggestion to restrict the study population to those born after 2003 to enhance the validity of our findings, this approach has an important limitation. Our datasets include hospital records only until 2020 in Sweden and 2016 in Denmark. Restricting the cohort to individuals born after 2003 (n=2 074 320) would result in a maximum follow-up period of 17 years. This smaller cohort and shorter follow-up period are unlikely to have a sufficient number of CVD cases in the group with maternal PCOS for well-powered analyses, even more so when studying subtypes of CVD.

Comment:

2. Also, is it unclear what's the stratification factor used in Table 4. What does "attained age" mean? The ages of attaining CVD or PCOS? And either way, what's the age used for those who did not have CVD or maternal PCOS? A more appropriate stratified analysis could be a Cox regression model stratified by the calendar year of birth and should be included as a sensitivity analysis.

Response:

We now clarify in the footnote of eTable 4 that the term "attained age" refers to the age of study participants at their exit from the cohort (i.e., having a CVD diagnosis, death, emigration, or the latest date in our datasets); i.e. attained age is equivalent to the age at the respective follow-up. This term is also referred to as "age as the time dimension" or "age as underlying time scale" or "age used as a left-truncation point". The purpose of using attained age as the underlying time scale in the Cox regression model, as well as in the stratified analysis by different attained age groups, was to address potential confounding by attained age, which cannot be handled by stratifying by calendar year of birth. This approach is clearly recommended for epidemiological studies where the outcome is strongly age dependent as in our case (*Breslow NE, et al. Multiplicative models and cohort analysis. J Am Statist Assoc 1983; 78: 1-12.*) as it provides the best possible adjustment for age.

However, we acknowledge that the calendar year of birth could also be a potential confounder in the association between maternal PCOS exposure and CVD outcomes in offspring and have therefore adjusted for it in our main analysis and in all our sub-analyses.

Reviewer #2 (Remarks to the Author):**Comment:**

1. The authors responses and modifications have improved the paper, however for reviewer 2 a few areas need further revision.

There are no references for the claims around higher nulliparity and lower fecundity. The literature shows that those with PCOS have similar nulliparity rates, albeit with OI or ART,

than those without PCOS. At most, the literature shows a 4% difference. Family size is also a little lower but largely similar. These reasons are inadequate and do not explain the poor capture rate of PCOS status. The NIH diagnosis point is valid., whereby a prevalence of around 8% vs 12% on Rotterdam, would see the proportion of cases captured in controls of at around 1 case for every 10 captured in the controls. This is ubiquitous in studies of this nature but specifically needs to be stated in terms of its impact here. The magnitude of the case load in the controls needs to be specified and recognised.

Response:

We appreciate the Reviewer's comments and acknowledge the need for further clarification. Please find below our response:

Our study population consists of women who had achieved pregnancy and had given birth over five decades, some of whom also had PCOS. Therefore, the prevalence of PCOS in our pregnancy cohort cannot be directly compared to the prevalence in the general population of women based on surveys in community samples.

We agree with the Reviewer that the literature does not suggest higher nulliparity rates among women with PCOS than those without and have revised possible claims about this. However, based on data from Table 1 (also shown below) we have noted that women without PCOS were more likely to have more births than those with PCOS. This finding may partially explain the lower prevalence of PCOS in our study, as women without PCOS contribute with more births to the denominator in the prevalence rate calculation than women without PCOS.

We have now included a reference (no. 44) in the manuscript to substantiate the well-established fact that PCOS is the most common cause of anovulatory infertility, as approximately 90-95% of anovulatory women seeking infertility treatment have PCOS. Furthermore, the table below shows that the proportion of women with PCOS undergoing ART is significantly higher than that of women without PCOS (17.1% vs. 2.9%). This supports our statement that "the clinical manifestation of PCOS involves subfertility and infertility."

Despite these possible explanations for the lower prevalence of PCOS in our study than in the general population of women, we agree with the Reviewer that it is likely that there is an underreporting of the PCOS diagnoses in our data due to the limited coverage of the patient registers. Women with milder PCOS symptoms, who did not seek medical care, or who were

diagnosed in private care, might have been classified as not having the disease. However, this underreporting possibly underestimated the strength of the studied association. In line with the Reviewer's suggestion, we have added the following note to the Discussion section:

“Although the positive predictive value of the PCOS diagnosis codes in the national patient registers was reported to be high (86%), it is likely that only women who sought medical care for clinically significant symptoms of PCOS or other medical conditions were diagnosed with PCOS; thus, the mothers with PCOS in our study may represent a less healthy group with more severe phenotypes. Women with milder PCOS symptoms, who did not seek medical care, or those who were diagnosed in private care, might have been misclassified as not having PCOS. Similarly, only data from inpatient care and/or from the Swedish MBR were available for the first part of the study period. We expect this underreporting to result in an underestimation of the association between maternal PCOS and the risk of CVD in the offspring.” (Please see page 17, line 1-11).

Table 1. The distributions of parity and assisted reproductive treatment among women without or with polycystic ovary syndrome

	Women without PCOS	Women with PCOS
	N (%)	N (%)
Parity		
1	2 945 493 (43.4)	28 453 (55.0)
2	2 496 503 (36.8)	16 924 (32.7)
≥3	1 345 984 (19.8)	6 346 (12.3)
Assisted reproductive treatment during the index pregnancy		
No	3 217 427 (97.1)	38 028 (82.9)
Yes	94 478 (2.9)	7 869 (17.1)

PCOS=polycystic ovary syndrome.

Comment:

2. CHD is almost always structural and is a totally different condition to metabolic CVD, IHD, hypertensive HD and stroke. Including CHD because it is related to the same organ as metabolic CVD and may be on the “same causal pathway” does not appear to be a clinically valid argument. Ideally the CHD would be excluded as the rationale provided does not make

clinical sense.

Response:

We appreciate the Reviewer's insights on this matter. According to several earlier studies, maternal PCOS is associated with an increased risk of congenital heart disease (CHD) in offspring (*Doherty DA, et al. Implications of Polycystic Ovary Syndrome for Pregnancy and for the Health of Offspring. Obstet Gynecol. 2015 Jun;125(6):1397-1406.*). Meta-analyses report that CHD is a major cause of CVD in childhood and early adulthood (*Wang T, et al. Congenital Heart Disease and Risk of Cardiovascular Disease: A Meta-Analysis of Cohort Studies. J Am Heart Assoc. 2019 May 21;8(10):e012030.*). Individuals with CHD, even after surgery, often have residual anatomic and hemodynamic abnormalities in the cardiovascular system, making them particularly vulnerable to ischemic heart disease, stroke, or heart failure (*Wang T, et al. Congenital Heart Disease and Risk of Cardiovascular Disease: A Meta-Analysis of Cohort Studies. J Am Heart Assoc. 2019 May 21;8(10):e012030.*). The suggested underlying mechanisms for the association between maternal PCOS and CVD risk in offspring involve CHD-related structural and functional changes in the offspring's heart and vessels, subsequently increasing the risk of several of the CVDs that we considered as outcome. We have now clarified this in the text and provided references to substantiate our hypothesis that CHD may be a mediator between maternal PCOS and CVD in the offspring: "The potential mediating role of CHD could be supported by previous findings that: 1) maternal PCOS is associated with an increased risk of CHD in offspring, and 2) CHD is a major cause of CVD in childhood and early adulthood due to its influence on anatomic and hemodynamic abnormalities in the cardiovascular system." (Please see page 16, line 7-10).

Observational epidemiologic studies commonly advise against conditioning on mediators of the causal link between exposure and outcome, as this can introduce bias (*Groenwold RHH, et al. To Adjust or Not to Adjust? When a "Confounder" Is Only Measured After Exposure. Epidemiology. 2021 Mar 1;32(2):194-201.*). In our case, conditioning on the mediator, CHD (either by adjusting for CHD or by restricting the analysis to individuals without CHD suggested by the reviewer), might lead to collider bias. This is particularly problematic if there are unmeasured common causes of both CHD and CVD that we cannot control for (see Figure 1 below).

Figure 1. Directed acyclic graph showing the association between maternal polycystic ovary syndrome (PCOS) and cardiovascular disease in the offspring (CVD) with congenital heart disease (CHD) as the potential mediator. The covariates (C) are a set of confounders with available information as listed above, and the arrow from C to CHD may be applicable for some but not all covariates. “U” is a set of unmeasured confounders, such as genetic predisposition or environmental risk factors. The dashed line indicates a false association caused by collider bias.

Comment:

3. BMI is inexorably linked to CVD and to PCOS. A comment in the limitations on the lack of BMI data is needed. This includes the low rate of obesity compared to current prevalence.

Response:

We address, among the limitations of the study, the concern that in the earlier years of our study period data on maternal body-mass index (BMI) were not recorded in the birth registers: “Further, data on maternal smoking, BMI and the use of ART before the index birth were not available for all years of our study period, which is why we could not control for these measures in our main analyses. However, adjustment for maternal smoking, BMI or the use of ART in sensitivity analyses among study participants with information on these variables did not substantially influence our estimates.” (Please see page 18, line 20-21; page 19, line 1-3).

We also mention, among the limitations of our study, that “Additionally, as there have been substantial changes in maternal age at childbirth and maternal BMI over the past five decades, the generalizability of the findings to children born in the current period is not clear.” (Please see page 19, line 12-14).

The offspring’s BMI may also be a mediator in the association between maternal PCOS and the risk of CVD in offspring. Unfortunately, we did not have data on the offspring’s BMI which prevents us from testing this hypothesis. We address this concern in the Discussion section: “Maternal PCOS may also increase the offspring’s risk of CVD by increasing the risk of obesity in the offspring, which in turn is a well-known cardiovascular risk factor. Regrettably, information on offspring’s BMI was not available in registers, preventing us from further investigating this hypothesis.” (Please see page 16, line 16-20).

Comment:

4. As noted above, number of children per women in PCOS will have marginal impact here and should be removed or deemphasised in the rationale for very low capture rates. The authors may wish to highlight that their work does align to similar prevalence capture in large population studies of this nature.

Response:

We have now deemphasized the argument about the potential impact of the number of births on the calculations of the prevalence of PCOS in our cohort. In the revised manuscript, we state: “In addition, women without PCOS in our cohort were more likely to contribute with more births to the denominator of the PCOS prevalence rate than those with a PCOS diagnosis, potentially leading to a lower rate of maternal PCOS in this birth cohort than the rate of PCOS in a cohort of women from the total population.” (Please see page 17, line 17-19).

We now also mention in the Discussion section that the observed prevalence of PCOS in our study aligns with findings from previous large population-based studies in Denmark and Sweden (*Schmidt AB, et al. Polycystic ovary syndrome and offspring risk of congenital heart defects: a nationwide cohort study. Hum Reprod. 2020 Oct 1;35(10):2348-2355; Roos N, et*

al. Risk of adverse pregnancy outcomes in women with polycystic ovary syndrome: population based cohort study. BMJ. 2011 Oct 13;343): “The prevalence of PCOS in our population is similar to that reported in previous Danish and Swedish register-based studies of a similar nature.” (Please see page 17, line 20-22).

Reviewer #3 (Remarks to the Author):

Comment:

I approve of the revisions. My only issue is that I think that the term "trend toward an association" for results that are not significant is valid. Although often said, I think trend should be reserved for changes over time, or explained as to what you are referring to as a trend.

Response:

Thank you for your approval. We have revised the sentence as follows: “...; the point estimate corresponding to the risk of stroke in offspring according to exposure to maternal PCOS was also increased, although the 95% CIs included one (Table 2).” (Please see page 11, lines 6-8).

REVIEWER COMMENTS

Reviewer #1 (Remarks to the Author):

Comment:

Although this revision has addressed some of my previous concerns, my major concern was not completely addressed. In particular, I don't understand why the authors could not conduct a sensitivity analysis by restricting the study sample to those whose birth year was after 2003. Although that'll result in a smaller sample size with shorter follow-up years, it'll still be a sample of about 2 million individuals. And even if the parameter estimate has a wider confidence interval due to much lower power, the magnitude of the estimated hazard ratio in this sensitivity analysis should be similar to the main analysis if the conclusion of this study is robust. Or another approach is to conduct stratified analysis, stratifying by year of birth (before 1990, 1990-2003, and after 2003) to see if the hazard ratios in different strata are relatively consistent across different calendar years of birth.

Response:

We appreciate the reviewer's suggestion and would be willing to conduct the suggested sensitivity or stratified analyses, but unfortunately we have technical obstacles to perform them. Access to researchers from Sweden to the data involving the Danish-Swedish cohort was in the frame of an interinstitutional agreement between Karolinska Institutet and Aarhus University. The current study was part of the first author's doctoral thesis defended earlier this year and the interinstitutional agreement was largely written for the period of her doctoral studies. As the contract is now expired, no one from the Swedish team has currently access to the data stored on a secure server at Statistics Denmark run by Aarhus University. Even within Aarhus University there is an ongoing process for an internal transition of the server between departments given a decrease and changes in employment among Danish team members. The study team is working on finding a solution to this data transfer between study centers and within Aarhus University, but the process of renewal of contracts and associated data transfer involves multiple institutions and is expected to take several months.

However, we believe that the proposed additional analysis would not substantially alter our main findings because of the following reasons:

1. Although restricting the cohort to individuals born after 2003 would provide a sample size of around 2 million study participants, this cohort would consist of a very young population, with an age at the end of follow-up ranging between 0-17 years. The power to detect a modest association between exposure and cardiovascular disease in such a young cohort may be limited, as cardiovascular diseases are very rare in this age group. Furthermore, findings from this cohort may not be generalizable to older populations.

2. We performed earlier a sensitivity analysis in which we restricted the study period to those born in Denmark after 1995 and those born in Sweden after 2001, i.e. the periods when the countries' patient registers included also specialised outpatient care (see Supplementary eTable 5). The risk estimates in analyses restricted to this subpopulation were similar to those for the entire cohort (Table 1). The cohort born in 2003 or later would to a large extent overlap with the cohort in these analyses, but be smaller, somewhat younger and have fewer cardiovascular diseases both overall and in the PCOS group. We would expect in the cohort born in 2003 or later a comparable point estimate and wider confidence intervals than those in the restricted analyses shown in the Table 1 below.

Table 1. Hazard ratios and 95% confidence intervals for overall cardiovascular diseases in offspring according to maternal polycystic ovary syndrome

	Number of events	Event rate, per 10 000 person-years	HR (95% CI) ^a
Overall population			
No PCOS	382 782	23.93	1.0 (Reference)
PCOS	1492	22.41	1.21 (1.15-1.27)
Restricted to offspring born in Denmark since 1995 and in Sweden since 2001			
No PCOS	49 500	15.51	1.0 (Reference)
PCOS	798	19.00	1.17 (1.09-1.25)

^a Adjusted for sex, country and calendar year of birth, maternal country of origin, parity, age, education, and marital status at the time of birth, hypertensive disorders, diabetes, and psychiatric disorders before or during the index pregnancy, and family history of cardiovascular disease.

3. From the reviewer's initial comment, i.e. "The severe underdiagnosis of PCOS in earlier years could substantially bias the estimates for the associations of interest. I would strongly recommend the authors to only focus on data after 2003 to ensure the validity of the conclusions of this study", we understand that one of the reviewer's concerns relates to the varying definition and validity of the PCOS diagnoses over time. Our exposure of interest, maternal PCOS, is defined based on a lifetime diagnosis. Therefore, restricting the analyses to study participants born in 2003 would not necessarily mean that the maternal PCOS diagnosis would be diagnosed according to the Rotterdam criteria and be recorded in the patient registers in 2003 or later. To address the concern that the diagnostic criteria of PCOS varies over time, we performed a stratified analysis according to the year of PCOS diagnosis (i.e., before 1990, 1990-2003, and after 2003). The risk estimates for these subgroups were comparable to each other and to the estimate in the overall sample; please see Supplementary eTable 5.

If the reviewer and editors argue that analyses restricted to those born after 2003 would significantly contribute to addressing the concerns raised in addition to (1) the previously performed analyses to consider confounding by age and calendar period and misclassification of exposure related to calendar period and (2) the earlier related revisions to the text based on comments from reviewer 1 and former reviewer 2, we can conduct them using the Swedish sub-cohort on a secure server in Sweden. However, please note that this would also require several months for data extraction, linkage, and analysis and in case of the Swedish sub-cohort the power would be even lower than in case of the analyses discussed above.

Reviewer #4 (Remarks to the Author):

Comment:

With regard to comment 2 re CHD, I would agree with the original reviewer that CHD is not on the causal pathway for CVD and I would have thought that those with CHD should be excluded. The mechanisms causing CVD in those with CHD would surely be different to those without CHD. However, the authors have substantiated their views to justify why those with CHD should not be excluded.

I believe the authors have satisfactorily addressed comments to reviewer 2.

Response:

Thank you.

REVIEWER COMMENTS

Reviewer #1 (Remarks to the Author):

Comment:

This revision has adequately addressed my previous concerns. I have no additional comments except that I suggest the authors moving Table 1 in the response letter to the supplementary tables. Also, they should add a few sentences discussing the results of sensitivity analysis (Table 1 in the response letter) in the Discussion section.

Response:

Thank you for the suggestions. The Table 1 from the previous response letter is part of the Supplementary Table 2. We have now extended the discussion about the results of this sensitivity analysis: "...Additionally, sensitivity analyses restricted to individuals born in Denmark after 1995 and in Sweden after 2001 yielded results consistent with those from the entire cohort." (Page 12, Line 21-22).